# *EchoDistill*: BIDIRECTIONAL CONCEPT DISTILLATION FOR ONE-STEP DIFFUSION PERSONALIZATION

## ABSTRACT

Recent advances in accelerating text-to-image (T2I) diffusion models have enabled the synthesis of high-fidelity images even in a single step. However, personalizing these models to incorporate novel concepts remains a challenge due to the limited capacity of one-step models to capture new concept distributions effectively. We propose a bidirectional concept distillation framework, *EchoDistill*, to enable one-step diffusion personalization (*1-SDP*). Our approach involves an *end-to-end* training process where a multi-step diffusion model (teacher) and a one-step diffusion model (student) are trained simultaneously. The concept is first *distilled* from the teacher model to the student, and then *echoed* back from the student to the teacher. During the *EchoDistill*, we share the text encoder between the two models to ensure consistent semantic understanding. Following this, the student model is optimized with adversarial losses to align with the real image distribution and with alignment losses to maintain consistency with the teacher's output. Furthermore, we introduce the bidirectional *echoing* refinement strategy, wherein the student model leverages its faster generation capability to feedback to the teacher model. This bidirectional concept distillation mechanism not only enhances the student ability to personalize novel concepts but also improves the generative quality of the teacher model. Our experiments demonstrate that this collaborative framework significantly outperforms existing personalization methods over the *1-SDP* setup, establishing a novel paradigm for rapid and effective personalization in T2I diffusion models.

## 1 INTRODUCTION

Recently, large-scale generative models (Ma et al., 2023a; Tu et al., 2024; Xing et al., 2024; Zhang et al., 2023a) dominate high-quality text-to-image (T2I) generation and have been widely applied in diverse downstream tasks (Hertz et al., 2023; Mou et al., 2024; Wang et al., 2023a; Zhang et al., 2023b). Among these applications, *personalized* text-to-image generation, also referred to as *new concept learning* (Chung et al., 2025; Kumari et al., 2023; Wu et al., 2025c), has emerged as a particularly important task. It involves adapting a T2I model to recognize and synthesize a novel concept from user-provided reference images. Recent T2I personalization methods (Gal et al., 2023a; Kumari et al., 2023; Ruiz et al., 2023a) generally adapt pretrained T2I models using few-shot reference images and bind the novel concept to a pseudo-token so that the adapted model can synthesize various renditions of the new concept guided by text prompts. Despite their success, the adapted T2I models still face a notable limitation which lies in their slow inference speed. To address inference inefficiency for T2I models, recent research has turned to distillation-based acceleration techniques (Liu et al., 2025; Sauer et al., 2024; Zheng et al., 2024; Xu et al., 2025). These techniques have matured considerably in the context of T2I diffusion models (Luo et al., 2023b; Salimans & Ho, 2022; Sauer et al., 2024), as we focus on in this paper. In general, training-based distillation methods aim to learn a fast student generator (Luo et al., 2023a; Sauer et al., 2024; Song et al., 2023; Zheng et al., 2024) from a multi-step T2I diffusion teacher model. Representative acceleration methods (Dao et al., 2024; Luo et al., 2023a; Sauer et al., 2024) achieve impressive acceleration by reducing the number of sampling steps to four or even fewer than one step for image generation.

However, existing personalization methods often overlook this critical requirement for few-step diffusion models. Directly applying conventional personalization techniques to these acceleration models frequently results in failure cases. As an example illustrated in Fig. 1, the *word-inversion* method Textual Inversion applied to the one-step diffusion model SDTurbo (Sauer et al., 2024) is unable to learn the textual concept tokens, which indicates that the one-step diffusion model struggles to

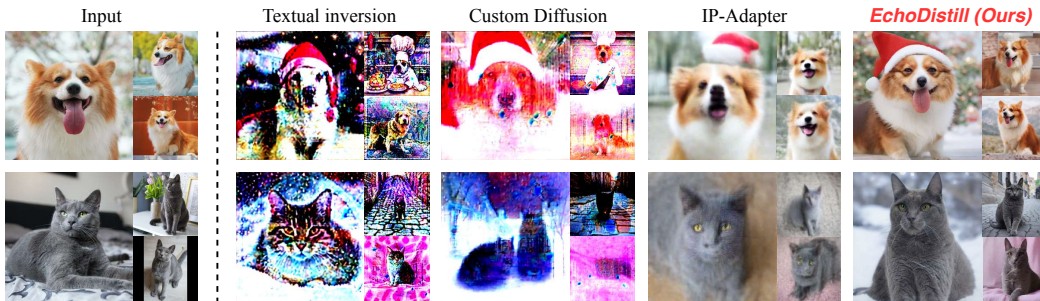

| Input | Textual inversion | Custom Diffusion | IP-Adapter | *EchoDistill (Ours)* |

Figure 1: Comparison with existing new concept learning methods for one-step personalization: Textual Inversion (Gal et al., 2023a) and Custom Diffusion (Kumari et al., 2023) for SDTurbo (Sauer et al., 2024), and IP-Adapter (Ye et al., 2024) for TCD (Zheng et al., 2024).

independently update the text encoder. This presents the first challenge in *1-SDP*: **1) Student inadaptability:** The student cannot learn text tokens independently and effectively. This issue is further exemplified in our experiments with the *weight-optimization* method Custom Diffusion (Kumari et al., 2023) in one-step models (Fig. 1), where jointly updating the text encoder and the diffusion backbone fails to improve performance and instead degrades generation quality. For existing *encoder-based* personalization methods, they struggle to generalize to one-step diffusion due to its unique architectural and optimization properties (IP-Adapter in Fig. 1). A naive way to leverage the distillation technique for the *1-SDP* problem is to first fine-tune a multi-step teacher model on the target concept, then use it to generate diverse samples to distill the one-step student. However, this results in two additional challenges: **2) Inefficiency:** The multi-step generation process and non-end-to-end teacher-student distillation will significantly slow down learning. **3) Teacher irreliability:** The teacher itself can also fail to capture certain concepts, limiting its effectiveness as a guiding signal for the student. These issues contribute to significant failure cases of current methods in concept personalization for one-step diffusion student models.

In this paper, we address the above challenges in *1-SDP* by introducing our *EchoDistill* framework. The *EchoDistill* jointly trains a multi-step T2I teacher model and a one-step student model in an end-to-end manner. The framework consists of two collaborative learning stages. In the first, *distillation* stage, the pretrained multi-step T2I teacher model concurrently learns the target concept while transferring knowledge to the student via concept distillation. In the second, *echoing* stage, the fast-generating student model produces images that are leveraged to further refine and enhance the teacher's generative performance. More specifically, to address the above challenges in *1-SDP*, we propose the following strategies along the *distillation stage*: **1) Shared text encoder (*STE*):** To ensure semantic consistency and improve knowledge transfer between models, the student directly inherits the text encoder from the teacher model; **2) End-to-end joint training (*E2E*):** Rather than relying on a sequential training paradigm, we adopt a unified optimization framework where the teacher and student models are trained simultaneously on the target concept. This promotes stable knowledge distillation and faster convergence. To support training, we employ two types of loss functions: alignment losses to ensure consistency with the teacher model, and adversarial losses to align the student's outputs with the real image distribution of the novel concept. **3) Echoing stage (*Echo*):** Following the first *distillation* stage, we further introduce a bidirectional refinement *echoing* stage to improve the performance. We exploit the student model's fast generation ability by using its high-quality outputs to reverse-guide the learning of the teacher and student models. After one-step personalization, our method *EchoDistill* is also able to achieve few-step (2-step, 4-step, etc.) customized generations as a bonus. To assess the performance of our method, *EchoDistill*, we compare it against several personalization approaches using a range of evaluation schemes from DreamBench (Ruiz et al., 2023a). The results, both qualitative and quantitative, highlight the superior effectiveness of our approach. To summarize, we make the following contributions:

- To the best of our knowledge, we are the *first* to identify and formalize the problem of one-step diffusion personalization, termed *1-SDP*, which significantly accelerates generation compared to conventional multi-step diffusion models while maintaining competitive image quality. We identify three core challenges in this setup: student inadaptability, inefficiency, and teacher irreliability.

- To tackle these challenges, we introduce a set of novel solutions: 1) a shared text encoder (*STE*) to mitigate student inadaptability by aligning semantic understanding between teacher and student; 2) a joint end-to-end framework (*E2E*) that personalizes the one-step student model alongside

the multi-step teacher model; and 3) an echoing stage (*Echo*), where the student's fast image generation is leveraged to refine and stabilize the teacher's output.

- We conduct extensive experiments on the DreamBench (Ruiz et al., 2023a) benchmark, evaluating both qualitative and quantitative performance. Our proposed *EchoDistill* outperforms existing personalization techniques under the *1-SDP* setting, demonstrating its ability to quickly adapt to novel concepts while preserving generation quality in one-step student models.

## 2 RELATED WORKS

Text-to-image personalization, also known as new concept learning, focuses on adapting a model to a user-provided novel concept using a few reference images. This technique has been extensively studied in T2I diffusion models (Butt et al., 2024; Gal et al., 2023a; Liu et al., 2023a; Ruiz et al., 2023a) and recently studies in the AR domain (Chung et al., 2025; Wu et al., 2025c).

**Tuning-based methods** (Li et al., 2025) leverage reference images of the same concept to fine-tune either the T2I diffusion model or its learnable embeddings. Depending on the optimization targets, these methods can be categorized into word-inversion and weight-optimization approaches.

*Word-inversion* methods focus on learning new concept tokens without modifying the parameters of generative models. Textual Inversion (Gal et al., 2023a) is a pioneering approach that introduces pseudo-words by performing personalization in the text embedding space. Other works (Agarwal et al., 2024; Dong et al., 2022; Voynov et al., 2023; Zhao et al., 2025) continually enable fine-grained and robust concept representation by employing designed loss functions to ensure that each token captures a distinct aspect of the reference images. While these methods maintain high semantic consistency by keeping the generative model frozen, they suffer from limited identity fidelity due to the compression of rich image features into the low-dimensional text embedding space.

*Weight-optimization* methods advance beyond token-level personalization by fine-tuning the model's internal weights, enabling richer and more faithful concept learning. One of the most prominent methods is DreamBooth (Ruiz et al., 2023a), which fine-tunes a pre-trained text-to-image diffusion model to associate a unique identifier with a target subject using just 3–5 reference images. Following that, several methods such as Custom Diffusion (CD) (Kumari et al., 2023) and Cones 2 (Liu et al., 2023a) propose optimizing only a subset of model parameters, significantly reducing both training time and memory consumption while preserving generation fidelity. Along similar lines, a variety of approaches (Chen et al., 2023; Gal et al., 2023b; Han et al., 2023; Zhang et al., 2023d) have emerged to improve visual quality and efficiency. In addition to partial weight tuning, recent works (Achlioptas et al., 2023; Xiang et al., 2023) introduce parameter-efficient strategies using Adapter modules, Low-Rank Adaptation (LoRA), or their variants, including Hyper-E4T (Arar et al., 2023), DisenBooth (Chen et al., 2024), etc.

**Tuning-Free methods** have proposed encoder-based alternatives that significantly reduce or eliminate the need for fine-tuning backbones by leveraging pre-trained image encoders. These methods (Li et al., 2024; Rowles et al., 2024; Shi et al., 2024; Wang et al., 2024a; Xiao et al., 2023; Ye et al., 2024) enable efficient concept learning by extracting informative features from reference images using models trained on large-scale, diverse datasets. Some of them also specify in human face generation (Cui et al., 2024; Guo et al., 2024; Li et al., 2024; Wang et al., 2024a; Wu et al., 2024). A recent advanced research is IP-Adapter (Ye et al., 2024), which utilizes the ViT image encoder from CLIP (Radford et al., 2021) to extract reference image features. These features are then integrated into the diffusion model's U-Net backbone through cross-attention mechanisms, resulting in more coherent and faithful renditions. While encoder-based methods are effective for general personalization from a single reference image, they are mainly tailored for large-scale, multi-step T2I models and require expensive retraining for each new backbone. Their limited ability to capture concept diversity (Li et al., 2025) and lack of adaptation to few-step architectures remain key limitations. Applying such pretrained encoders or adapters to one-step diffusion models yields poor concept fidelity and image quality, highlighting a critical gap in current research.

## 3 METHODOLOGY

In this section, we begin by introducing the preliminaries in Section 3.1, followed by revealing the key challenges of this *1-SDP* setup in Section 3.2. We then present our framework, *EchoDistill*, which incorporates three novel techniques designed to address these challenges in Section 3.3.

### 3.1 PRELIMINARIES

**Latent Diffusion Models.** LDM (Rombach et al., 2022) is the most widely applied T2I diffusion models and the distillation teacher model for current few-step diffusion models (Luo et al., 2023a; Sauer et al., 2024). It is conditioned on textual input $\tau(\mathcal{P})$, where $\tau$ is the text encoder and $\mathcal{P}$ is the prompt. The backbone $\epsilon_\theta^{tc}$ is a conditional UNet (Ronneberger et al., 2015) which predicts the added noise. After predicting the noise, diverse schedulers (Lu et al., 2022; Song et al., 2021) are used to denoise. Here, we use SD2.1 as the teacher model $\epsilon_\theta^{tc}$ as in T2I acceleration approaches (Nguyen & Tran, 2024; Sauer et al., 2024).

**One-Step Diffusion Model.** To accelerate diffusion inference, various methods distill the sampling steps $T_{tc} = [1, T]$ of the teacher into few student anchor steps (NFEs*) $T_{st} = \{v_1, \ldots, v_n\}$ where $n$ is typically set to 1, 2, or 4. Specifically, a *one-step* diffusion model $\mathcal{G}_{st}$ aims to transform a noise $x_T \sim \mathcal{N}(0, 1)$ directly into an image without iterative denoising steps, hence we denote this noise to image process as $x_0^{st} = \mathcal{G}_\phi^{st}(x_T, T, \mathcal{C})$. In this paper, we build on the one-step SD-Turbo (Sauer et al., 2024) as the student model $\mathcal{G}_\phi^{st}$ for new concept learning under the *1-SDP* setup.

### 3.2 CORE CHALLENGES IN *1-SDP*

As previously illustrated qualitatively in Section 1 and supported quantitatively in Section 4, conventional T2I personalization methods fail to learn new concepts in one-step diffusion models (such as SDTurbo (Sauer et al., 2024)). We identify three main challenges in this *1-SDP* setting.

**Student Inadaptability.** We begin by applying the *word-inversion* Textual Inversion (Gal et al., 2023a) to the one-step diffusion model SD-Turbo(Sauer et al., 2024). Observing from Fig. 1 and Table 1, this naive adaptation fails to capture or reproduce the target concept, revealing a key limitation: one-step diffusion models cannot be effectively personalized by text encoder tuning alone, indicating the need for additional supervision or architectural changes. We further evaluate *weight-optimization* methods Custom Diffusion on SD-Turbo. This not only fails to enhance performance but also degrades image quality and concept fidelity, as proven in Fig. 1. These findings suggest that excessive flexibility in updating the backbone of few-step models may disrupt the generative prior. We hypothesize that this limitation arises from the inherent differences in the distillation objectives. Traditional diffusion models distill the entire generative process, preserving detailed noise-to-image mappings. In contrast, few-step models are typically trained using distribution alignment losses (Poole et al., 2023; Wang et al., 2023b) rather than reconstructing individual denoising trajectories. As a result, applying conventional diffusion losses during personalization leads to ineffective learning and poor visual fidelity.

**Teacher Irreliability and Inefficiency.** A naive approach to tackle the *1-SDP* problem is a two-stage distillation strategy, which we refer to as the *teacher-first* distillation paradigm. In this setup, a multi-step teacher model is first fine-tuned on the target concept, and then used to generate diverse supervision samples for training the one-step student model. However, this paradigm faces two fundamental limitations: *1) Inefficiency:* The pipeline requires the teacher to first complete concept learning before it can supervise the student. Furthermore, the supervision involves multi-step generation (e.g., using 25 or 50 NFEs), making the process slow and computationally expensive. *2) Teacher Irreliability:* The teacher model may struggle to accurately learn certain visual concepts, particularly under sparse supervision, as can be seen from the Custom Diffusion baseline (SD2.1) in Table 1 and Fig. 3. When this occurs, the generated samples used for distillation are suboptimal or even misleading, thereby degrading the student model's performance due to poor supervision. Results for the teacher-first paradigm and additional discussions are provided in Appendix. H.

### 3.3 *EchoDistill*: BIDIRECTIONAL CONCEPT DISTILLATION

Based on the above observations, we propose the first end-to-end bidirectional concept distillation framework, termed *EchoDistill*, specifically designed for personalizing one-step diffusion models (*1-SDP*). Unlike the above teacher-first paradigm that sequentially updates the teacher (multi-step diffusion model) and then distills knowledge to the student (one-step diffusion model), our approach adopts a gradual *distillation* and *echoing* bidirectional concept distillation strategy, where the student learns progressively in tandem with the teacher. In addition, the student provides feedback to the teacher during the following *echoing* stage. To ensure memory efficiency and prevent overfitting, we adopt the lightweight adaptation strategy from Custom Diffusion (Kumari et al., 2023), updating

---

*NFEs denote the number of function evaluations, from the view of diffusion ODE trajectories.

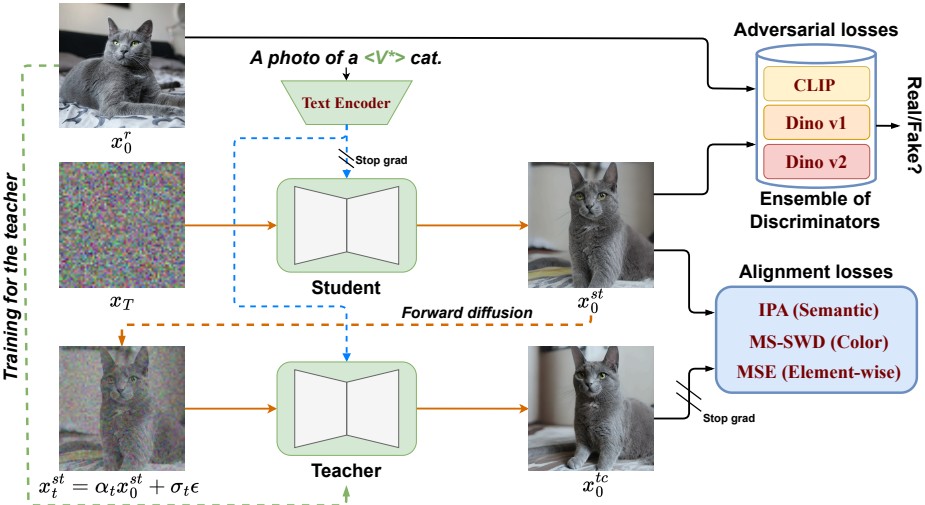

Figure 2: Overview of *EchoDistill*. The student and teacher jointly learn the new concept with a shared text encoder. The teacher learns from real images $x_0^r$ (green line), and the text encoder is updated accordingly. The student is optimized with two objectives (gold line): an adversarial loss to match real data distribution and alignment losses to match the denoised outputs of the teacher. The discriminators are trained to distinguish between the student's outputs and real images.

only the key and value projections in both teacher and student models. To tackle the aforementioned challenges, *EchoDistill* includes three targeted strategies to address the above challenges.

### 3.3.1 DISTILLATION STAGE: SHARED TEXT ENCODER AND END-TO-END DISTILLATION

To address the first challenge—student adaptability—we propose the use of a shared text encoder (*STE*) between the teacher and student models. Motivated by the observation that the student can benefit from consistent semantic grounding, we directly inherit the teacher's text encoder for the student during distillation. This design ensures a unified language-vision alignment across both models. To facilitate effective training under this shared encoder and also to address the second challenge of inefficient training, we adopt a bidirectional gradual distillation strategy. In this scheme, the student model is progressively trained alongside the teacher. To maintain memory efficiency and mitigate overfitting during distillation, we adopt the lightweight adaptation strategy from Custom Diffusion, updating only the key and value projections in both teacher and student networks.

Our training procedure is illustrated in Fig. 2 and the detailed algorithm pipeline is in Appendix. F. *EchoDistill* consists of three steps in each iteration. *First*, the real image $x_0^r$ is fed into the teacher model. The *teacher* is trained following the Custom Diffusion paradigm, where both the text encoder and the UNet are optimized using the noise prediction loss $\mathcal{L}_{\text{rec}}$:

$$\mathcal{L}_{rec} = \mathbb{E}_{x_0^r, y, t, \epsilon \sim \mathcal{N}(0,1)} \| \epsilon - \epsilon_\theta^{tc}(x_t, t, \tau(\mathcal{P})) \|_2^2 \tag{1}$$

*Second*, the *student* receives a random noise $x_T \sim \mathcal{N}(0,1)$ input and generates an output $x_0^{st} = \mathcal{G}^{st}(x_T, T, \mathcal{C})$. This output is guided by a combination of two objectives: *(1) alignment losses* between the student and teacher, and *(2) adversarial losses* between the student and real images. For the alignment objective, $x_0^{st}$ is passed through the teacher's forward diffusion process to obtain a noisy version $x_t^{st} = \alpha_t x_0^{st} + \sigma_t \epsilon, \epsilon \sim \mathcal{N}(0,1)$, which is then denoised by the teacher to yield the predicted $x_0^{tc}$. This $x_0^{tc}$ is detached via a stop-gradient operation and serves as the supervision target for computing the alignment losses against $x_0^{st}$. For the *adversarial* objective, the student is optimized to fool an ensemble of discriminators, which are trained to distinguish the student-generated image $x_0^{st}$ from real images $x_0^r$. *Third*, the discriminators are optimized to enhance their discriminative performance. The detailed loss functions are defined as follows.

**Alignment losses** encourage the student-generated images to be semantically consistent with those from the teacher model, capturing both low-level pixel details and high-level perceptual alignment. It is composed of three components:

- Identity Feature Loss, adapted from IP-Adapter (Ye et al., 2024) (IPA), extracts identity-preserving features from the image space $x_0$ using a CLIP image encoder followed by a projection network.

Here the $x_0^{tc}$ is the estimated teacher image, computed from the teacher prediction $x_0^{tc} = \alpha_t^{-1/2} \cdot \{x_t - [(1 - \alpha_t) \cdot \epsilon_t^{tc} / (1 - \bar{\alpha}_t)^{1/2}]\}$ while the $x_0^{st}$ is the image from student direct generation $\mathcal{G}^{st}(x_T, T, \mathcal{C})$. This loss is computed as cosine similarity:

$$\mathcal{L}_{\text{id}}(x_0^{st}, x_0^{tc}) = 1 - \cos\left(\text{IPA}(x_0^{st}), \text{IPA}(x_0^{tc})\right) \tag{2}$$

- MSE Loss minimizes the distance between student and teacher latent representations:

$$\mathcal{L}_{\text{mse}}(x_0^{st}, x_0^{tc}) = \left\| x_0^{st} - x_0^{tc} \right\|_2^2 \tag{3}$$

- Multi-Scale Sliced Wasserstein Distance (He et al., 2024), compares multi-scale feature distributions in the image space to align structural and color information. This loss is proposed to alleviate the unstable color distribution during the distillation and defined as:

$$\mathcal{L}_{\text{swd}}(x_0^{st}, x_0^{tc}) = \text{MS-SWD}\left(x_0^{st}, x_0^{tc}\right) \tag{4}$$

The full alignment loss scales these components by weighting factors and a time-dependent term:

$$\mathcal{L}_{\text{align}} = c(t) \cdot \left[\lambda_{\text{id}} \cdot \mathcal{L}_{\text{id}}(x_0^{st}, x_0^{tc}) + \lambda_{\text{mse}} \cdot \mathcal{L}_{\text{mse}}(x_0^{st}, x_0^{tc}) + \lambda_{\text{ms}} \cdot \mathcal{L}_{\text{swd}}(x_0^{st}, x_0^{tc})\right] \tag{5}$$

Inspired by ADD (Sauer et al., 2024), we introduce a timestep-dependent exponential weighting factor $c(t) = \alpha(t)$, where $t$ denotes the randomly sampled timestep in the teacher's noising–denoising process and $\alpha(t)$ is the same as defined in DDPM (Ho et al., 2020). At higher noise levels (i.e., larger $t$), the teacher's predictions become increasingly unreliable, and the $c(t)$ is accordingly decreased. This design helps stabilize the student's training by reducing the influence of noisy supervision.

**Adversarial losses** are designed to reduce the distribution gap between student-generated outputs and real concept images. Specifically, we ensemble multiple discriminators (Chan et al., 2022; Kumari et al., 2022), each operating from a different semantic perspective, to improve training stability and achieve better results. We employ $K = 3$ discriminators, each using a different pretrained backbone—DINOv1, DINOv2, and CLIP—as feature extractors. Each backbone is followed by a two-layer trainable projection head to distinguish between real and generated images, while the feature extractors remain frozen during training. The adversarial loss for the student is defined as:

$$\mathcal{L}_{\text{GAN}}^G = \sum_{k=1}^{K} \lambda_k \cdot \mathbb{E}_{x_0^{st}}\left[-\log(D_k(x_0^{st}))\right] \tag{6}$$

where $D_k$ denotes the $k$-th discriminator (based on DINOv1 (Caron et al., 2021), DINOv2 (Oquab et al., 2023) and CLIP (Radford et al., 2021)), and $x_0^{st}$ is the student output image. Denoting $x_0^r$ as the real image, the discriminator loss is defined as:

$$\mathcal{L}_{\text{GAN}}^D = -\sum_{k=1}^{K} \left[\mathbb{E}_{x_0^r}\left[\log D_k(x_0^r)\right] + \mathbb{E}_{x_0^{st}}\left[\log(1 - D_k(x_0^{st}))\right]\right] \tag{7}$$

In summary, the integration of a shared text encoder (*STE*), alignment losses and adversarial loss collectively enhances the adaptability and generalization capacity of the student model within the first distillation stage of our *EchoDistill* framework. It is worth *noting that* these loss formulations are specifically designed for the one-step student model, whose fast image generation allows efficient access to final outputs. In contrast, applying such losses to the multi-step teacher is *impractical* due to the computational cost of obtaining real image outputs across *iterative* denoising steps.

### 3.3.2 ECHOING STAGE: STUDENT IMPROVES THE TEACHER

We interpret the one-step student model as a GAN-like generator, and hypothesize that it can benefit from aligning with the few-shot target *data distribution* via adversarial training. Specifically, incorporating a adversarial loss (Sauer et al., 2024; Wang et al., 2024b; 2018b; Yin et al., 2024) as in Eq. 6 and Eq. 7 helps the student model generate samples that better match the *distribution* of concept images, even enabling it to *outperform the teacher model* in terms of *visual qualities*, as demonstrated in Table 1. This finding also aligns with insights from ADD (Sauer et al., 2024), which emphasizes the critical role of the discriminator loss in boosting generative *fidelity*.

Building on this insight, we propose an additional *echoing* stage, which leverages the student model's rapid generation capability to provide constructive feedback to both itself and the teacher model. The echoing stage mirrors the distillation stage, with the only difference being the definition of the real trainig examples $x_0^r$. Specifically, we replace real images with randomly generated samples from the updated student model after the first-stage distillation: $\hat{x}_0^r = \mathcal{G}^{st}(x_T, T, \mathcal{C})$. The training objectives and update rules for both the student and teacher models remain unchanged from the initial distillation stage. The motivation behind this design is further discussed in Appendix F.

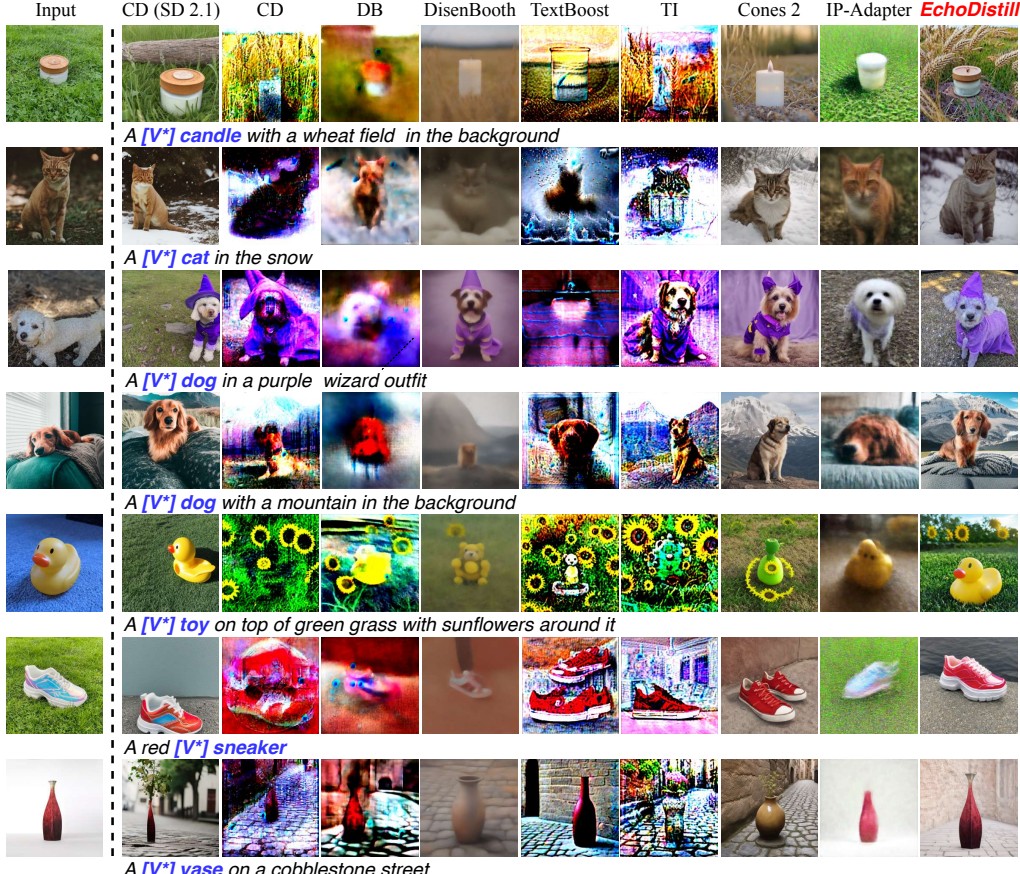

Figure 3: Our method *EchoDistill* (last column) compared with existing methods applied to the *1-SDP* setup with SDTurbo (Sauer et al., 2024) as the one-step diffusion backbone. One representative concept image is shown on the left-most column.

## 4 EXPERIMENTS

### 4.1 EXPERIMENTAL SETUPS

**Comparison methods and evaluation metrics.** We compare our method *EchoDistill* with the following T2I personalization approaches: 1) word-inversion: Textual Inversion (Gal et al., 2023a), Cones 2 (Liu et al., 2023a); 2) optimization-based: Custom Diffusion (Kumari et al., 2023), DreamBooth (Ruiz et al., 2023a), DisenBooth (Chen et al., 2024), TextBoost (Park et al., 2024); 3) Encoder-based: IP-Adapter (Ye et al., 2024). For Disenbooth and Cones2 approach, the reference images were taken from training dataset in TextBoost. For Custom Diffusion, we also include a baseline variant using SD2.1 (Rombach et al., 2022) as the backbone, which also serves as the base model for SDTurbo. For IP-Adapter, due to the absence of an available implementation compatible with the one-step SDTurbo model, we adopt the TCD-based practice (Zheng et al., 2024) for comparison. We follow the default configurations in their papers or open-source implementations. In the experiments, we evaluate our method on the DreamBooth (Ruiz et al., 2023a) dataset, which contains 30 distinct concepts for personalized learning. To measure the alignment between generated images and textual prompts, we employ the CLIP-T score. Additionally, we assess cosine visual similarity between generated images and reference images using DINO (Caron et al., 2021) and CLIP-I (Radford et al., 2021) metrics, following standard practices in prior works (Kumari et al., 2023; Park et al., 2024). More details of these comparisons and experiments are in the Appendix. G.

**Implementation Details.** During training of our method *EchoDistill*, the teacher model employs a sampling schedule with 1000 denoising steps (NFEs), while the student model performs denoising in a single step. For the hyperparameters, the loss weights in Eq. 5 and Eq. 6 are set such that $\lambda_{ms} = 0.1$, while all other weights are set to 1.0. The model is trained with a learning rate of $2 \times 10^{-5}$ and a batch size of 2. All experiments are conducted on a single NVIDIA A40 GPU.

Table 1: Quantitative comparisons with existing text-to-image (T2I) personalization methods.

| Methods | Model | Train NFEs | Infer NFEs | CLIP-T | CLIP-I | DINO |
|---|---|---|---|---|---|---|
| Custom Diffusion | SD 2.1 | 1000 | 25 | 0.269 | 0.752 | 0.519 |
| Custom Diffusion | SD Turbo | 1 | 1 | 0.205 | 0.518 | 0.058 |
| Textual Inversion | SD Turbo | 1 | 1 | 0.252 | 0.564 | 0.166 |
| Cones 2 | SD Turbo | 1 | 1 | **0.273** | 0.619 | 0.204 |
| DreamBooth | SD Turbo | 1 | 1 | 0.188 | 0.536 | 0.111 |
| TextBoost | SD Turbo | 1 | 1 | 0.217 | 0.570 | 0.167 |
| DisenBooth | SD Turbo | 1 | 1 | 0.251 | 0.564 | 0.231 |
| IP-Adapter | TCD | 1 | 1 | 0.204 | 0.628 | 0.325 |
| *EchoDistill* | SD Turbo | 1 | 1 | 0.252 | **0.783** | **0.637** |

Table 2: Ablation Study (Part 1).

| Methods | CLIP-T | CLIP-I | DINO |
|---|---|---|---|
| ***Ablate Components*** | | | |
| w/o teacher | 0.240 | 0.719 | 0.505 |
| w/o discriminators | 0.200 | 0.566 | 0.105 |
| Full model | **0.252** | **0.783** | **0.637** |
| ***Ablate Infer-NFEs*** | | | |
| 1 step | 0.252 | **0.783** | **0.637** |
| 2 steps | 0.255 | **0.783** | 0.635 |
| 4 steps | **0.256** | 0.772 | 0.610 |

Table 3: Ablation Study (Part 2).

| Methods | | CLIP-T | CLIP-I | DINO |
|---|---|---|---|---|
| ***Ablate 1-Step Backbone*** | | | | |
| Hyper-SD1.5 | | 0.211 | 0.709 | 0.463 |
| SD-turbo | | **0.252** | **0.783** | **0.637** |
| ***Ablate Echo Stage*** | | | | |
| Teacher | Before Echo | **0.269** | 0.752 | 0.519 |
| | After Echo | 0.265 | **0.764** | **0.571** |
| Student | Before Echo | **0.252** | 0.783 | 0.637 |
| | After Echo | 0.236 | **0.798** | **0.673** |

## 4.2 EXPERIMENTAL RESULTS

**Qualitative results.** The main qualitative comparisons are presented in Fig. 3. Among the seven baseline methods, Custom Diffusion, DreamBooth, TextBoost, and Textual Inversion fail to perform effective denoising or learn target concepts under the one-step inference setting. DisenBooth and Cones2 struggle to capture precise concepts. Although IP-Adapter preserves some identity consistency, its results are often blurry, misaligned with the prompts, and affected by the reference image background. In contrast, our proposed method, *EchoDistill*, achieves both precise concept learning and strong semantic alignment between the generated images and input texts.

**Quantitative results.** The detailed numeric results are presented in Table 1. *EchoDistill* maintains the text-image alignment quality competitive to the baselines, as evidenced by the CLIP-T Score. In terms of image similarity (CLIP-I, DINO scores), *EchoDistill* significantly outperforms the one-step or multi-step based methods. We *note that* the CLIP-T is not that relevant in T2I personalization since it is computing the alignment with the generated image with the input text (without the conditional token), and therefore does not capture the alignment with the input concept images or the intended new concept. Therefore the CLIP-I and DINO are more *convincing* to demonstrate the effectiveness of personalization methods: on these scores our method shows the best performance. Moreover, *EchoDistill* even surpasses the Custom Diffusion (SD2.1) in the first row, further supporting our argument that the teacher model is not fully reliable under the *1-SDP* setup. Additionally, the low CLIP-I and DINO scores of the Textual Inversion baseline highlight the student's inability to independently learn the concept without effective supervision. These results collectively validate the key challenges we identified in adapting concept learning to the *1-SDP* setup.

## 4.3 ABLATION STUDY

**Main components.** Table 2 presents the analysis of removing key components from *EchoDistill*, namely the teacher model and the discriminators. As shown, removing the teacher or the discriminators leads to a noticeable decline across all evaluation metrics. These results indicate that both components play a crucial role in effectively learning new concepts. Qualitative ablation results are shown in Fig. 4-(c). Our full model generates high-quality and semantically consistent outputs. In contrast, omitting the teacher results in images that lack fine details specific to the target concept. When the discriminators are removed, the outputs are significantly more noisy. This degradation likely stems from the teacher's use of a single denoising step, which produces $x_0$ predictions with residual noise that are subsequently propagated to the student model during training.

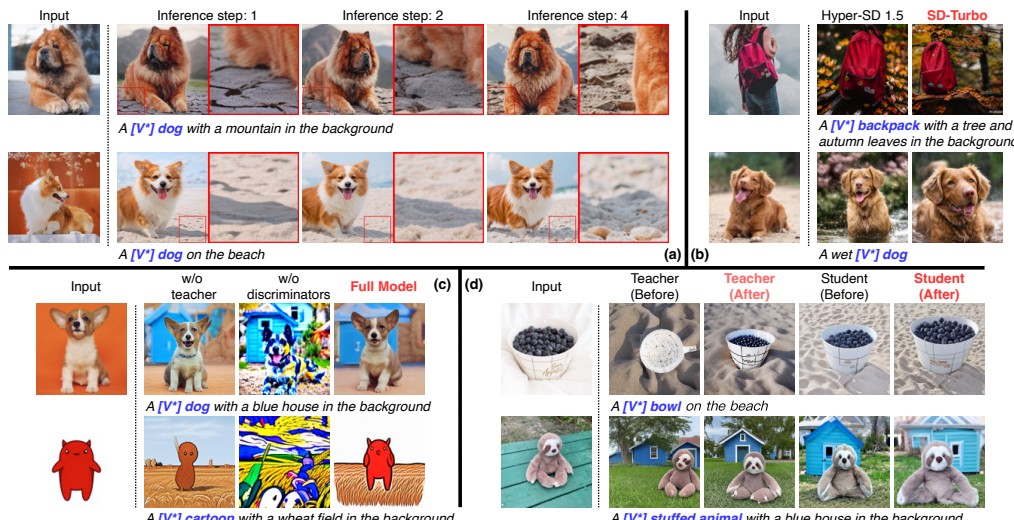

Figure 4: (a) Abating the Infer-NFEs; (b) Ablating the one-step diffusion backbone; (c) Ablating the teacher and discriminators; (d) Ablating the echoing stage.

**Inference Steps.** Although *EchoDistill* is trained for a 1-step setting, we further evaluate its performance for 2-step and 4-step denoising, using the same trained model without re-training. As reported in Table 2, CLIP-T scores show slight improvements with additional steps, whereas CLIP-I and DINO scores exhibit marginal declines. Overall, the variations across quantitative metrics remain minimal. The qualitative comparisons in Fig. 4-(a) reveal more perceptible differences. Notably, increasing the number of inference steps enhances image fidelity, especially by producing richer background details and finer textures. This generalizability emerges as a beneficial byproduct of the training process in *EchoDistill*.

**1-Step Backbones.** We perform a backbone ablation study to assess the adaptability of our method to alternative one-step backbones. In particular, we replace the student model with Hyper-SD1.5 and adjust the teacher model accordingly to SD1.5. The quantitative results are summarized in Table 3. Our findings indicate that this alternative backbone yields inferior performance compared to the SDTurbo backbone. However, as illustrated in Fig.4-(b), Hyper-SD1.5 still remains capable of generating reasonable outputs in such cases. Notably, our choice of SDTurbo as the primary backbone is motivated by two main factors: it is one of the few one-step models capable of generating high-quality images, and its distillation-based training process is well aligned with our framework, which likely contributes to its superior performance over Hyper-SD1.5 when used as the backbone.

**Echoing Stage.** In Table 3, we compare the student and teacher performance after the *echoing* stage. The teacher model exhibits significant improvements in CLIP-I and DINO scores, while CLIP-T scores experience a slight decline. These results suggest that the student's output can effectively enhance the performance of both teacher and student models, particularly in terms of identity and visual similarity. Qualitative examples in Fig. 4-(d) further support this observation: when the teacher model struggles to learn certain concepts, leveraging the student's output as an additional supervisory signal enables the teacher to better capture and reproduce those challenging concepts.

## 5 CONCLUSIONS

In this paper, we introduced the novel task of one-step diffusion personalization (*1-SDP*), a significant step toward bridging the gap between fast generative inference and concept-personalized image synthesis. We identified three major challenges that prevent conventional personalization methods from being directly applicable to one-step diffusion models. To overcome these limitations, we proposed a unified framework, *EchoDistill*, which jointly trains a multi-step teacher and a one-step student through an end-to-end distillation process and a bidirectional echoing stage. By leveraging a shared text encoder, end-to-end optimization, and student-guided echoing stage, *EchoDistill* enables effective adaptation to new visual concepts. Extensive experiments on the DreamBench benchmark confirm that *EchoDistill* consistently outperforms existing personalization approaches, setting a new foundation for rapid and reliable concept learning in diffusion-based generation.

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

APPENDIX

## A    BROADER IMPACTS

Our method, *EchoDistill*, enables efficient and high-fidelity personalized image generation, which holds significant potential for a range of creative applications, including design, education, and virtual content creation. By reducing the need for extensive data and computation, *EchoDistill* democratizes access to advanced generative tools, empowering users with minimal resources to produce customized visual content. However, as with other powerful generative models, our approach also introduces potential risks. These include the unauthorized generation of content, impersonation, and the creation of misleading or harmful imagery. We acknowledge these risks and stress the importance of deploying appropriate safeguards, such as content moderation, usage auditing, and user authentication mechanisms, particularly in real-world applications. We advocate for the responsible use of this technology and encourage the research community and stakeholders to collaborate on developing ethical guidelines and technical solutions to mitigate potential misuse.

## B    LIMITATIONS.

Our proposed method, *EchoDistill*, marks an initial step toward enabling one-step diffusion models to learn novel concepts efficiently. While the experimental results are promising, several limitations remain and warrant further investigation: *(1) Training efficiency:* The current training pipeline is hindered by the computational overhead introduced by the multi-discriminator architecture. Optimizing or rethinking this component could significantly improve training speed. *(2) Limited one-shot personalization:* The discriminator's reliance on multiple reference samples to model the underlying data distribution makes true one-shot personalization challenging. Designing a more robust discriminator or alternative mechanisms to enable faithful learning from a single image remains an open problem. *(3) Training instability:* As with many GAN-based methods, our approach may exhibit instability across runs, particularly for challenging concepts, where achieving optimal results may require a few trials. Enhancing training stability remains a promising direction for future work. We leave these challenges as compelling avenues for future research, aiming to build upon this initial framework to support broader generalization, compositionality, and efficiency.

## C    CODE RELEASE AND REPRODUCIBILITY STATEMENT

We provide a lightweight testing script of *EchoDistill* at the anonymous repository https://anonymous.4open.science/r/EchoDistill_Anonymous-D5C5/, which enables users to experience the inference process of our pretrained one-step model. The complete codebase, including training scripts and model checkpoints, will be released upon the acceptance.

## D    ETHICAL AND LLM STATEMENTS

We acknowledge the potential ethical implications of deploying generative models, including concerns related to privacy, data misuse, and the propagation of biases. All models used in this paper are publicly available, and we will release the modified codes to enable reproduction of our results. We also emphasize the potential misuse of customization approaches in generating misinformation, and we strongly encourage and support their responsible usage. Regarding the use of LLMs, we clarify that in this work they were only minimally employed, specifically for correcting grammatical errors.

## E    OVERVIEW OF TEXT-TO-IMAGE PERSONALIZATION METHODS

In this section, we present a comprehensive comparison of representative text-to-image personalization methods, expanding upon the overview introduced in $\lambda$-Eclipse (Patel et al., 2024). Table 4 provides an extended summary that systematically contrasts these approaches across several key dimensions, including support for single- or multi-subject personalization, training-free versus training-based paradigms, number of input images required, inference efficiency, etc.

Table 4: We provide an overview of representative text-to-image personalization methods by extending the summary introduced in $\lambda$-Eclipse (Patel et al., 2024). The base models listed correspond to those used in their original papers. For a fair comparison with the highlighted methods in our study, we re-implemented and adapted all approaches using the same base model configuration as described in our main paper. ChilloutMix is a community-contributed variant of the Stable Diffusion model (Rombach et al., 2022). Infinity refers to a variant of the Text-to-Image VAR model (Tian et al., 2024), while LlamaGen denotes a text-to-image auto-regressive (AR) model (Sun et al., 2024a).

| Method | Multi-Subject | Tuning-Free | Base Model | Input Images | Inference Steps | Note |
|---|---|---|---|---|---|---|
| Textual Inversion (Gal et al., 2023a) | ✗ | ✗ | SDv1.4 | Few-Shot | Multi-Step | Word-Inversion |
| P+ (Voynov et al., 2023) | ✗ | ✗ | SDv1.4 | Few-Shot | Multi-Step | Word-Inversion |
| ProsPect (Zhang et al., 2023c) | ✗ | ✗ | SDv1.4 | 1-Shot | Multi-Step | Word-Inversion |
| MATTE (Agarwal et al., 2024) | ✗ | ✗ | SDv1.4 | 1-Shot | Multi-Step | Word-Inversion |
| Cones 2 (Liu et al., 2023b) | ✓ | ✗ | SDv2.1 | Few-Shot | Multi-Step | Word-Inversion |
| DreamBooth (Ruiz et al., 2023a) | ✗ | ✗ | SDv1.4 | Few-Shot | Multi-Step | |
| ClassDiffusion (Huang et al., 2025a) | ✗ | ✗ | SDv1.5 | Few-Shot | Multi-Step | |
| DisenBooth (Chen et al., 2024) | ✗ | ✗ | SDv2.1 | 1-shot | Multi-Step | |
| CatVersion (Zhao et al., 2025) | ✗ | ✗ | SDv1.5 | Few-Shot | Multi-Step | |
| AttnDreamBooth (Pang et al., 2024) | ✗ | ✗ | SDv2.1 | 1-shot | Multi-Step | |
| ViCo (Tumanyan et al., 2023) | ✗ | ✗ | SDv1.4 | Few-Shot | Multi-Step | |
| TextBoost (Park et al., 2024) | ✗ | ✗ | SDv1.5 | 1-shot | Multi-Step | |
| NeTI (Alaluf et al., 2023) | ✗ | ✗ | SDv1.4 | Few-Shot | Multi-Step | |
| HyperDreamBooth (Ruiz et al., 2023b) | ✗ | ✗ | SDv1.5 | 1-shot | Multi-Step | |
| E4T (Gal et al., 2023b) | ✗ | ✗ | SD | 1-shot | Multi-Step | |
| Hyper-E4T (Arar et al., 2023) | ✗ | ✗ | SD | 1-shot | Multi-Step | |
| ARBooth (Chung et al., 2025) | ✗ | ✗ | Infinity (Han et al., 2025) | Few-Shot | Multi-Step | |
| Proxy-Tuning (Wu et al., 2025c) | ✗ | ✗ | LlamaGen (Sun et al., 2024a) | Few-Shot | Multi-Step | |
| Continual Diffusion (Smith et al., 2023) | ✓ | ✗ | SD | Few-Shot | Multi-Step | |
| Perfusion (Tewel et al., 2023) | ✓ | ✗ | SDv1.5 | Few-Shot | Multi-Step | |
| Custom Diffusion (Kumari et al., 2023) | ✓ | ✗ | SDv1.4 | Few-Shot | Multi-Step | |
| Cones (Liu et al., 2023a) | ✓ | ✗ | SDv1.4 | 1-shot | Multi-Step | |
| SVDiff (Han et al., 2023) | ✓ | ✗ | SDv1.5 | Few-Shot | Multi-Step | |
| FreeCustom (Ding et al., 2024) | ✓ | ✗ | SDv1.5 | 1-Shot | Multi-Step | |
| Mix-of-Show (Gu et al., 2024) | ✓ | ✗ | Chilloutmix | Few-Shot | Multi-Step | |
| LoRACLR (Simsar et al., 2025) | ✓ | ✗ | Chilloutmix | Few-Shot | Multi-Step | |
| Orthogonal (Po et al., 2024) | ✓ | ✗ | Chilloutmix | Few-Shot | Multi-Step | |
| OMG (Kong et al., 2024) | ✓ | ✗ | SDXL | Few-Shot | Multi-Step | |
| Zip-LoRA (Shah et al., 2023) | ✓ | ✗ | SDXL | Few-Shot | Multi-Step | |
| Break-A-Scene (Avrahami et al., 2023) | ✓ | ✗ | SDv2.1 | 1-shot | Multi-Step | |
| TokenVerse (Garibi et al., 2025) | ✓ | ✗ | Flux (Labs, 2024) | 1-shot | Multi-Step | |
| *EchoDistill* (Ours) | ✗ | ✗ | SDTurbo (Sauer et al., 2024) | Few-Shot | 1-step | |
| PhotoMaker (Li et al., 2024) | ✗ | ✓ | SDXL | 1-shot | Multi-Step | Human Face |
| ConsistentID (Huang et al., 2024) | ✗ | ✓ | SDv1.5 | 1-shot | Multi-Step | Human Face |
| InstantID (Wang et al., 2024a) | ✗ | ✓ | SDXL | 1-shot | Multi-Step | Human Face |
| Profusion (Zhou et al., 2023) | ✗ | ✓ | SDv2 | 1-shot | Multi-Step | Human Face |
| PuLID (Guo et al., 2024) | ✗ | ✓ | SDXL | 1-shot | Multi-Step | Human Face |
| Infinite-ID (Wu et al., 2024) | ✗ | ✓ | SDXL | 1-shot | Multi-Step | Human Face |
| LCM-Lookahead (Gal et al., 2024) | ✗ | ✓ | SDXL | 1-shot | Multi-Step | Human Face |
| InfiniteYou (Jiang et al., 2025) | ✗ | ✓ | Flux (Labs, 2024) | 1-shot | Multi-Step | Human Face |
| IP-Adapter (Ye et al., 2024) | ✗ | ✓ | SDv1.5 | 1-shot | Multi-Step | |
| ELITE (Wei et al., 2023) | ✗ | ✓ | SDv1.4 | 1-shot | Multi-Step | |
| UMM-Diffusion (Ma et al., 2023b) | ✗ | ✓ | SDv1.5 | 1-shot | Multi-Step | |
| InstantBooth (Shi et al., 2024) | ✗ | ✓ | SDv1.4 | Few-Shot | Multi-Step | |
| BLIP-Diffusion (Li et al., 2023) | ✗ | ✓ | SDv1.5 | 1-shot | Multi-Step | |
| JeDi (Zeng et al., 2024) | ✗ | ✓ | SDv1.4 | Few-Shot | Multi-Step | |
| Re-Imagen (Chen et al., 2022) | ✗ | ✓ | Imagen (Saharia et al., 2022) | 1-shot | Multi-Step | |
| SuTi (Chen et al., 2023) | ✗ | ✓ | Imagen (Saharia et al., 2022) | Few-Shot | Multi-Step | |
| Taming (Jia et al., 2023) | ✗ | ✓ | Imagen (Saharia et al., 2022) | 1-shot | Multi-Step | |
| Kosmos-G (Pan et al., 2024) | ✓ | ✓ | SDv1.5 | 1-shot | Multi-Step | |
| SSR-Encoder (Zhang et al., 2024) | ✓ | ✓ | SDv1.5 | 1-shot | Multi-Step | |
| $\lambda$-Eclipse (Patel et al., 2024) | ✓ | ✓ | Kandinsky (Arkhipkin et al., 2023) | 1-shot | Multi-Step | |
| FastComposer (Xiao et al., 2023) | ✓ | ✓ | SDv1.5 | 1-shot | Multi-Step | |
| Subject-Diffusion (Ma et al., 2024) | ✓ | ✓ | SDv2.1 | 1-shot | Multi-Step | |
| RMCC (Huang et al., 2025b) | ✓ | ✓ | SDXL | 1-shot | Multi-Step | |
| Emu2 (Sun et al., 2024b) | ✓ | ✓ | SDXL | 1-shot | Multi-Step | |
| MS-Diffusion (Wang et al., 2025) | ✓ | ✓ | SDXL | 1-shot | Multi-Step | |

---

**Algorithm 1** Training Pipeline of One-step Personalization in *EchoDistill*

---

1: **Input:** Real image dataset $\mathcal{D} = \{(x_0^r, y)\}$; teacher model $\epsilon_\theta^{tc}$; student model $\mathcal{G}^{st}$; discriminators $\{\mathcal{D}_k\}_{k=1}^K$; text encoder $\tau(\cdot)$; diffusion steps $T$; noise schedule functions $\{\alpha_t, \sigma_t\}$; weighting functions $c(t)$, $\lambda_{\text{id}}, \lambda_{\text{mse}}, \lambda_{\text{ms}}, \{\lambda_k\}_{k=1}^K$
2: **for** each training iteration **do**
3:      Sample real image and prompt: $(x_0^r, y) \sim \mathcal{D}$
4:      Random noise $\epsilon \sim \mathcal{N}(0, 1)$
5:      Encode prompt: $\mathcal{C} \leftarrow \tau(y)$
6:      **Step 1: Teacher training**
7:      Sample timestep $t \sim \mathcal{U}(1, T)$
8:      Generate noisy input: $x_t = \alpha_t x_0^r + \sigma_t \epsilon$
9:      Predict noise: $\hat{\epsilon} \leftarrow \epsilon_\theta^{tc}(x_t, t, \mathcal{C})$
10:      Compute loss: $\mathcal{L}_{\text{rec}} = \|\epsilon - \hat{\epsilon}\|_2^2$
11:      Update teacher model and the text encoder using $\mathcal{L}_{\text{rec}}$
12:      **Step 2: Student training**
13:      Sample latent: $x_T \sim \mathcal{N}(0, 1)$
14:      Generate image: $x_0^{\text{st}} \leftarrow \mathcal{G}^{st}(x_T, T, \text{stopgrad}(\mathcal{C}))$         $\triangleright$ stopgrad$(\cdot)$ denotes stop-gradient
15:      *// Alignment loss*
16:      Forward diffuse: $x_t^{\text{st}} = \alpha_t x_0^{\text{st}} + \sigma_t \epsilon$
17:      Denoise: $x_0^{\text{tc}} \leftarrow \text{stopgrad}(\epsilon_\theta^{tc}(x_t^{\text{st}}, t, \mathcal{C}))$
18:      Compute alignment loss:
        $\mathcal{L}_{\text{align}} = c(t) \cdot [\lambda_{\text{id}} \cdot \mathcal{L}_{\text{id}}(x_0^{st}, x_0^{tc}) + \lambda_{\text{mse}} \cdot \mathcal{L}_{\text{mse}}(x^s, x^t) + \lambda_{\text{ms}} \cdot \mathcal{L}_{\text{swd}}(x_0^{st}, x_0^{tc})]$
19:      *// Adversarial loss*
20:      Compute adversarial loss: $\mathcal{L}_{\text{GAN}}^G = \sum_{k=1}^K \lambda_k \cdot \mathbb{E}_{x_0^{st}}[-\log(D_k(x_0^{st}))]$
21:      Update student model using: $\mathcal{L}_{\text{st}} = \mathcal{L}_{\text{align}} + \mathcal{L}_{\text{GAN}}^G$
22:      **Step 3: Discriminator training**
23:      **for** each discriminator $\mathcal{D}_k$ **do**
24:         Compute the discriminator loss:
        $\mathcal{L}_{\text{GAN}}^{D_k} = -\left[\mathbb{E}_{x_0^r}[\log D_k(x_0^r)] + \mathbb{E}_{x_0^{st}}[\log(1 - D_k(\text{stopgrad}(x_0^{st})))]\right]$
25:         Update $\mathcal{D}_k$ using: $\mathcal{L}_{\text{GAN}}^{D_k}$
26:      **end for**
27: **end for**
28: **Output:** Trained teacher model $\epsilon_\theta^{tc}$, student model $\mathcal{G}^{st}$, and text encoder $\tau$

---

# F    ALGORITHMIC DESCRIPTION OF *EchoDistill* TRAINING

**Distillation Stage.** The full training procedure of one-step personalization in *EchoDistill* is described in Section 3.3.1 of the main paper. For completeness, Algorithm 1 provides the detailed step-by-step implementation. Each iteration consists of three steps: *(1)* the teacher model and text encoder are jointly optimized via the noise prediction loss $\mathcal{L}_{\text{rec}}$ following the Custom Diffusion paradigm; *(2)* training of the student model through a combination of alignment losses with the teacher's outputs and adversarial losses against real image data; and *(3)* updating the discriminators to improve their capacity to differentiate between real and synthesized samples.

**Echoing Stage.** During the echoing stage, the training procedure remains identical to the previous phase, except that the training examples are replaced with one-step inference samples generated by the student model. This design is beneficial in multiple ways. *(1)* In the initial distillation stage, the availability of real images is limited, and this constrained data scale impedes the training of the teacher model. By contrast, the student model exhibits the ability to learn data distributions from few images. That is a capability endowed by the discriminator, as validated in few-shot GAN frameworks such as TransferGAN(Wang et al., 2018a) and MineGAN(Wang et al., 2020). Consequently, during the echoing stage, our objective is to sample from the data distribution learned by the student model, using these samples as training examples to enhance the teacher model's performance. *(2)* Notably, the 1-step diffusion student model learns distributions distinct from those acquired by the teacher model. Similar observation can be found in ADD(Sauer et al., 2024), where the discriminator loss primarily shapes the data distribution of the student model, while the distillation loss facilitates

Table 5: Quantitative comparisons with existing text-to-image (T2I) personalization methods. NFEs indicates the number of function evaluations.

| Methods | Model | Train NFEs | Inference NFEs | CLIP-T↑ | CLIP-I↑ | DINO↑ | Train Time (s) | Inference Time (s) | iterations |
|---|---|---|---|---|---|---|---|---|---|
| | SD 2.1 | 1000 | 25 | 0.264 | 0.761 | 0.555 | 345 | 2.73 | 1000 |
| | SD Turbo | 1000 | 1 | 0.207 | 0.530 | 0.097 | 541 | 0.22 | 1000 |
| | SD Turbo | 1000 | 4 | 0.257 | 0.597 | 0.235 | 541 | 0.54 | 1000 |
| Custom diffusion | SD Turbo | 1000 | 25 | 0.276 | 0.647 | 0.337 | 541 | 1.57 | 1000 |
| | SD Turbo | 1 | 1 | 0.205 | 0.518 | 0.058 | 543 | 0.22 | 1000 |
| | SD Turbo | 1 | 4 | 0.246 | 0.556 | 0.109 | 543 | 0.53 | 1000 |
| | SD Turbo | 4 | 1 | 0.206 | 0.532 | 0.105 | 543 | 0.23 | 1000 |
| | SD Turbo | 4 | 4 | 0.258 | 0.600 | 0.244 | 543 | 0.54 | 1000 |
| Textual Inversion | SD Turbo | 1 | 1 | 0.252 | 0.564 | 0.166 | 2269 | 0.13 | 4000 |
| Cones 2 | SD Turbo | 1 | 1 | 0.273 | 0.619 | 0.204 | 2446 | 0.51 | 4000 |
| DreamBooth | SD Turbo | 1 | 1 | 0.188 | 0.536 | 0.111 | 281 | 0.14 | 1000 |
| TextBoost | SD Turbo | 1 | 1 | 0.217 | 0.570 | 0.167 | 64 | 0.15 | 500 |
| DisenBooth | SD Turbo | 1 | 1 | 0.251 | 0.564 | 0.231 | 905 | 0.18 | 2000 |
| Lora | SD Turbo | 1 | 1 | 0.212 | 0.585 | 0.160 | 141 | 0.15 | 800 |
| IP-Adapter | TCD | / | 1 | 0.204 | 0.628 | 0.325 | / | 0.39 | / |
| OminiControl | Flux | / | 1 | **0.279** | 0.727 | 0.455 | / | 2.48 | / |
| *EchoDistill* | SD Turbo | 1 | 1 | 0.252 | **0.783** | **0.637** | 3137 | 0.18 | 1000 |

convergence and enhances conceptual alignment with the teacher's outputs. Supporting evidence for this ablation study can be found in Table 8. *(3)* By shifting the training examples from few real image inputs to images generated by the 1-step student model, the teacher model is enabled to learn from the student's distribution, a distribution partially shaped by the discriminator's design and characterized by image features not inherently present in the teacher model, thereby yielding beneficial effects.

# G  ADDITIONAL RESULTS ON METHOD COMPARISON

In this section, we present additional quantitative and qualitative results to further validate the effectiveness and efficiency of our proposed method. Table 5 extends the comparisons from the main paper by reporting both training and inference time (in seconds), along with the number of optimization iterations required by each method. These results underscore the efficiency of our approach as a one-step diffusion model, achieving image generation in just *0.18 seconds* per instance during inference. Figures 6 through 9 provide additional qualitative comparisons against representative baseline methods. Each figure presents a concept reference image (left) followed by results from various approaches. Our method, *EchoDistill*, consistently delivers superior visual fidelity while maintaining rapid inference, highlighting its practical advantages for real-time or resource-constrained applications in one-step diffusion-based image generation (*1-SDP*).

**Discussion on runtime cost.** About the training time, this limitation is inherent to optimization-based customization approaches, which universally require additional runtime computation when encountering novel concepts. Conversely, existing optimization-free methods, including encoder-based frameworks (IP-Adatper (Ye et al., 2024), DreamO (Mou et al., 2025), Xverse (Chen et al., 2025), UNO (Wu et al., 2025b), InfiniteYou (Jiang et al., 2025), etc.) and unified models (BAGEL (Deng et al., 2025), GPT-4o (Achiam et al., 2023), OmniGen2 (Wu et al., 2025a), etc.), demand extensive datasets for training. Furthermore, no encoder-based or unified model to date fully supports few-step (or even one-step) diffusion models. This leaves the integration of one-step models' speed advantages with the versatile capabilities of unified models as an underexplored research direction. In this work, we aim to be the first to investigate the realization of one-step personalization via an optimization-based approach, with optimization-free alternatives designated as future work.

**Comparison with Flux+OminiControl.** We further compare *EchoDistill* with the recent Flux (Labs, 2024) model combined with the OminiControl (Tan et al., 2025). It is important to note that Omini-Control is trained on large-scale datasets similar to IP-Adapter, which makes the evaluation against our method not entirely equitable. The evaluation is conducted on the DreamBooth dataset under the 1-step setup, and the results are summarized in the lower part of Table 5. As shown, Flux+OminiControl outperforms other baselines reported in the table; however, it remains signif-

Table 6: Additional results for alternative designs on the DreamBooth dataset.

| Method | CLIP-T ↑ | CLIP-I ↑ | DINO ↑ |
|---|---|---|---|
| Teacher-first | **0.266** | 0.725 | 0.503 |
| Remove STE | 0.242 | 0.742 | 0.551 |
| Ours (*EchoDistill*) | 0.252 | **0.783** | **0.637** |

Table 7: User study for diverse methods.

| Method | Preference Rate (%) |
|---|---|
| Custom Diffusion (Kumari et al., 2023) | 32.26% |
| Cones 2 (Liu et al., 2023a) | 0.48% |
| DisenBooth (Chen et al., 2024) | 0.36% |
| *EchoDistill* | 66.90% |

icantly inferior to our proposed *EchoDistill*. This performance gap can be attributed to the weaker generation capability of Flux constrained to 1-step inference.

## H    ADDITIONAL DISCUSSIONS FOR ALTERNATIVE DESIGNS

We further compare our method with several alternative designs in order to clarify the motivation and validity of our proposed framework. All experiments in this section are performed on the Dream-Booth dataset.

**Teacher-first paradigm.** In this design, the teacher is first trained, and the teacher-generated samples for the target concept (with varying text prompts) are directly used as supervised training data for the student. The identity loss (Eq. 2 in the main paper) is applied between the teacher-generated samples and the student outputs, allowing the student to learn identity-related features from the teacher model. As shown in Table 6, this design performs worse than our proposed approach. Moreover, it suffers from several inherent limitations: (1) Computational overhead: teacher inference requires multiple steps, which is inefficient; (2) Teacher irreliability: as discussed in the main paper, the teacher does not always successfully learn the target concepts; (3) Limited image diversity: the generated images consistently feature highly similar visual appearances; and (4) Performance ceiling: the student's performance is inherently bounded by the capabilities of the teacher.

In addition, we also attempted to directly apply VSD (Wang et al., 2023b), SDS (Poole et al., 2023), and MSE losses to distill teacher-learned concepts into the student model under this paradigm. However, we observed that this approach was insufficient for transferring the teacher's personalization capabilities to the student.

**Discussion on feed-forward customization methods.** Beyond the teacher-first paradigm, an alternative direction is to build on feed-forward customization methods such as SynCD (Kumari et al., 2025) and JeDi (Zeng et al., 2024), and then distill these models into a few-step diffusion framework. However, most existing distillation techniques for diffusion models are primarily designed to align the data distributions of few-step models with those of their teacher models. This emphasis stems from the inherent difficulty that few-step diffusion models face in replicating the full denoising trajectory of their teacher. As a result, subtle discrepancies in concept-specific details are often introduced, as observed in prior works such as AYF (Sabour et al., 2025), ADD (Sauer et al., 2024), and LCM (Luo et al., 2023a).

**Remove STE.** We further investigate the effect of sharing the text encoder between the teacher and student models. Removing the shared text encoder (STE) results in a clear performance drop, demonstrating that STE not only simplifies the training framework but also improves learning efficiency.

Table 8: Ablation study of our method *EchoDistill*.

| Methods | CLIP Score | CLIP-I | DINO |
|---|---|---|---|
| Full model | **0.252** | **0.783** | **0.637** |
| w/o the teacher | 0.240 | 0.719 | 0.505 |
| w/o all the discriminators | 0.200 | 0.566 | 0.105 |
| w/o Identity Feature Loss | 0.248 | 0.769 | 0.618 |
| w/o MSE loss | 0.242 | 0.739 | 0.528 |
| w/o MS-SWD loss | 0.249 | 0.754 | 0.553 |
| w/o the discriminator of the Dino v1 | 0.231 | 0.689 | 0.441 |
| w/o the discriminator of the Dino v2 | 0.246 | 0.736 | 0.534 |
| w/o the discriminator of the Clip | 0.227 | 0.678 | 0.409 |

## I    USER STUDY

To evaluate alignment with human preferences, we conducted a user study involving 15 participants, yielding 840 preference annotations per method. In each trial, participants were presented with a set of generated images and instructed to "select the best image from each group, considering both text-image alignment and object identity consistency." The methods evaluated in our study include Custom Diffusion, Cones 2, and DisenBooth, which demonstrate superior performance compared to other baseline approaches based on both qualitative and quantitative experimental results. As summarized in Table 7, our approach, *EchoDistill*, significantly outperformed all baselines—receiving at least 34% more user votes than the second-best method. These findings underscore the strong alignment between *EchoDistill* 's outputs and human perceptual judgments.

## J    EXTENDED ABLATION STUDY

In Section 4.3 of the main paper, we explore the contributions of key components in *EchoDistill*, namely the teacher model and discriminators. A more detailed ablation study is presented in Table 8, wherein individual loss terms and discriminators are systematically removed. The results indicate that omitting the ID loss, latent MSE loss, or MSSWD loss causes notable performance degradation, particularly reflected in reduced DINO scores, underscoring their critical role in maintaining alignment with the teacher model. Furthermore, removal of any single discriminator leads to more pronounced declines across all evaluation metrics. Collectively, these findings demonstrate the complementary nature of the various loss functions and discriminators in improving generation fidelity and semantic consistency. Qualitative comparisons provided in Fig. 10 further illustrate the visual impact of removing each component. Beyond the general decline in generation quality, we observe that omission of certain components can induce training instability or divergence for specific concepts.

## K    1-SHOT PERFORMANCE.

We further evaluate our method under a one-shot supervision setting, wherein only a single image is utilized for training. As summarized in Table 9, performance declines across all evaluation metrics relative to the few-shot scenario. This degradation is anticipated, given that our approach is not explicitly optimized for one-shot learning, and the scarcity of supervisory data increases the likelihood of training instability. Qualitative results illustrated in Fig. 5 demonstrate that, although one-shot training can yield visually plausible outputs, the generated images occasionally lack fine-grained details corresponding to the novel concept.

Table 9: One-shot performance of our *EchoDistill*.

| Methods | CLIP-T | CLIP-I | DINO |
|---------|--------|--------|------|
| One-shot | 0.231 | 0.713 | 0.470 |
| Few-shot | **0.252** | **0.783** | **0.637** |

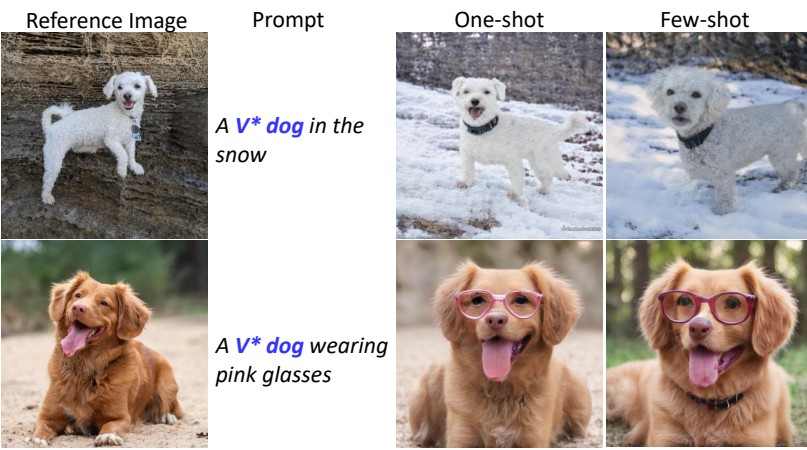

Figure 5: The qualitative results of the 1-shot performance.

## L    RESULTS ON THE CUSTOMCONCEPT101 DATASET

We further evaluate our method on the CustomConcept101 dataset (Kumari et al., 2023). Qualitative results, presented in Fig. 11 and Fig. 12, demonstrate that our approach generalizes effectively across a diverse set of concepts and prompt types, consistently generating high-quality outputs.

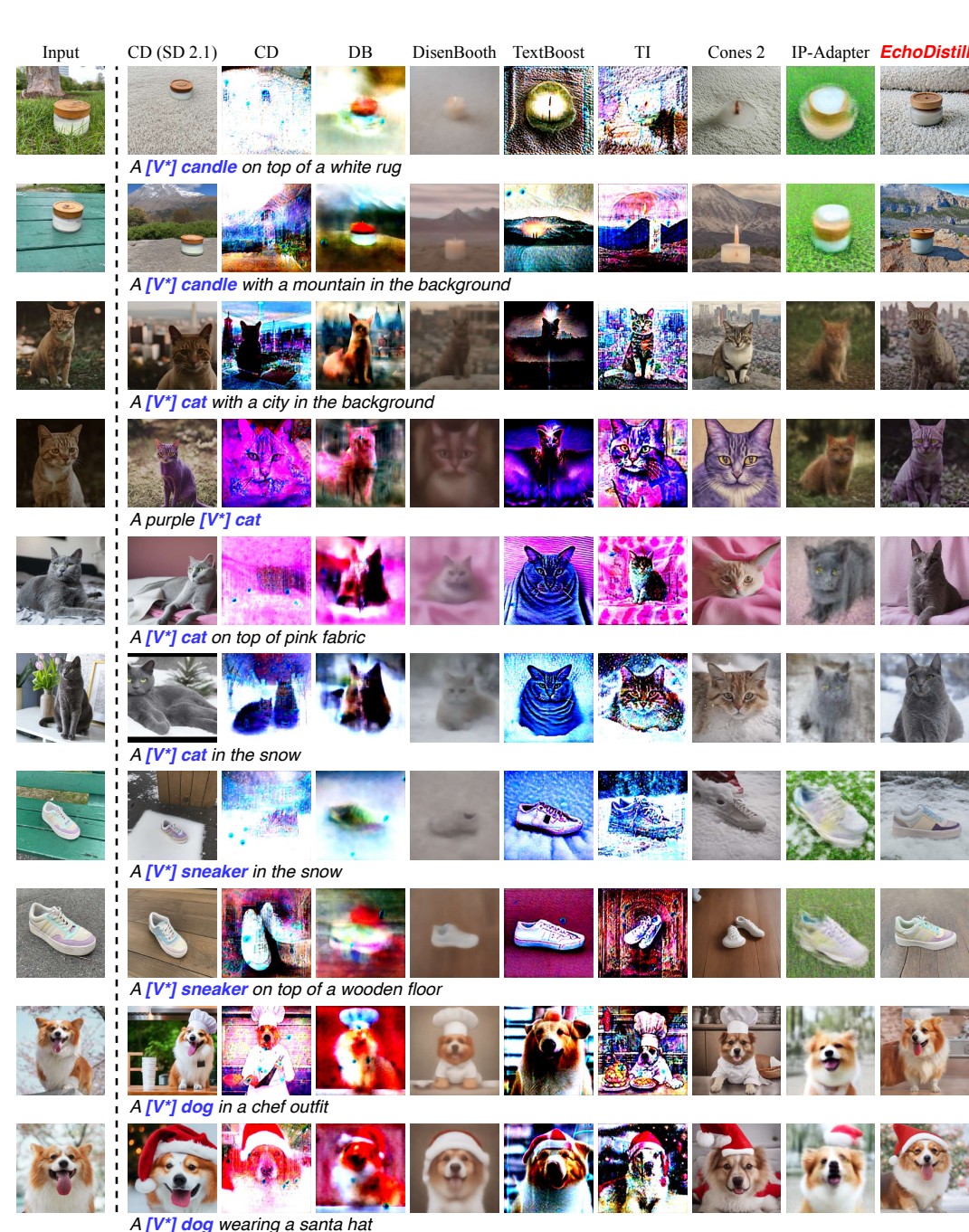

Figure 6: Our method *EchoDistill* (last column) compared with existing methods applied to the *1-SDP* setup with SDTurbo (Sauer et al., 2024) as the one-step diffusion backbone. One representative concept image is shown on the left-most column. (Part 1)

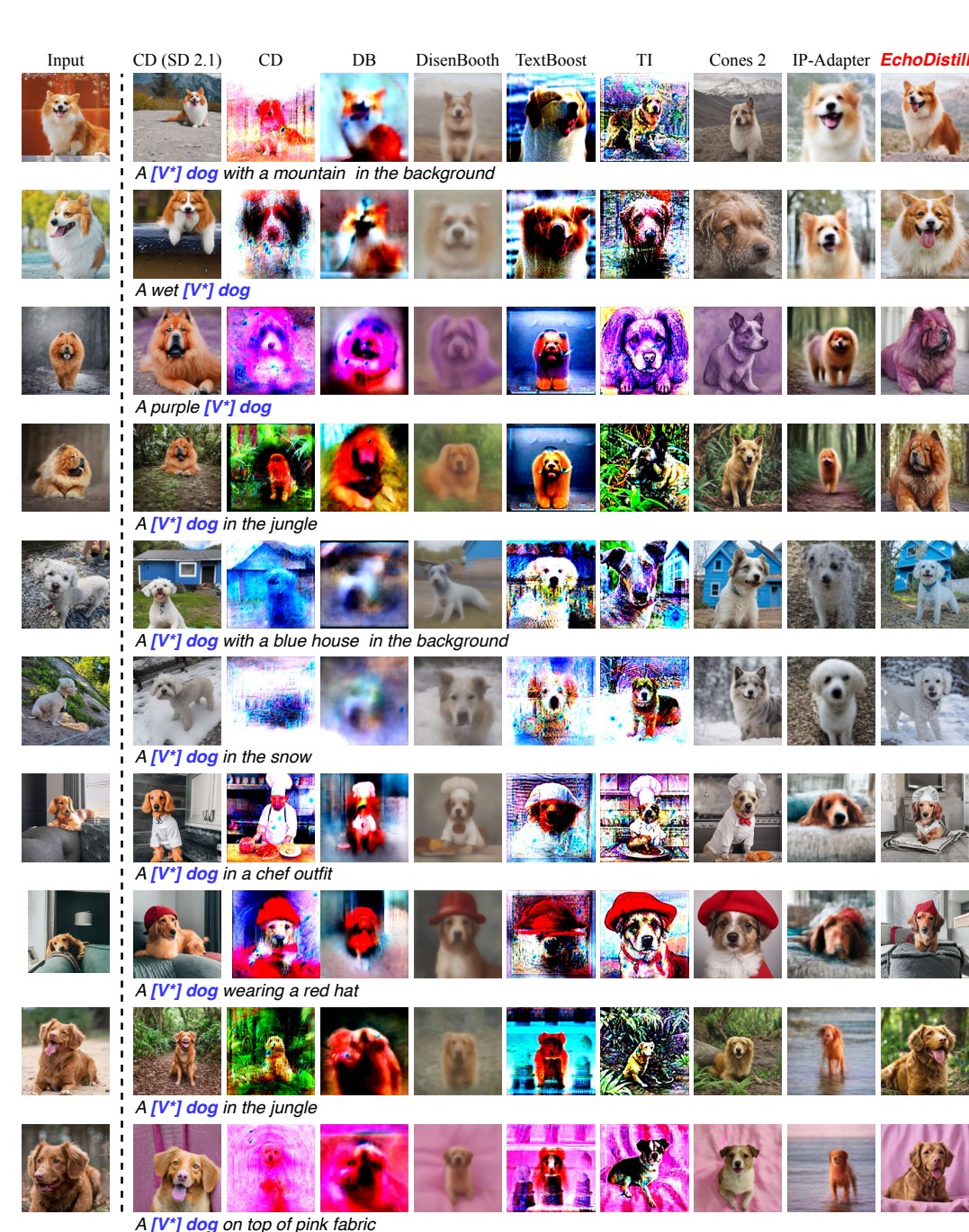

Figure 7: Our method *EchoDistill* (last column) compared with existing methods applied to the *1-SDP* setup with SDTurbo (Sauer et al., 2024) as the one-step diffusion backbone. One representative concept image is shown on the left-most column. (Part 2)

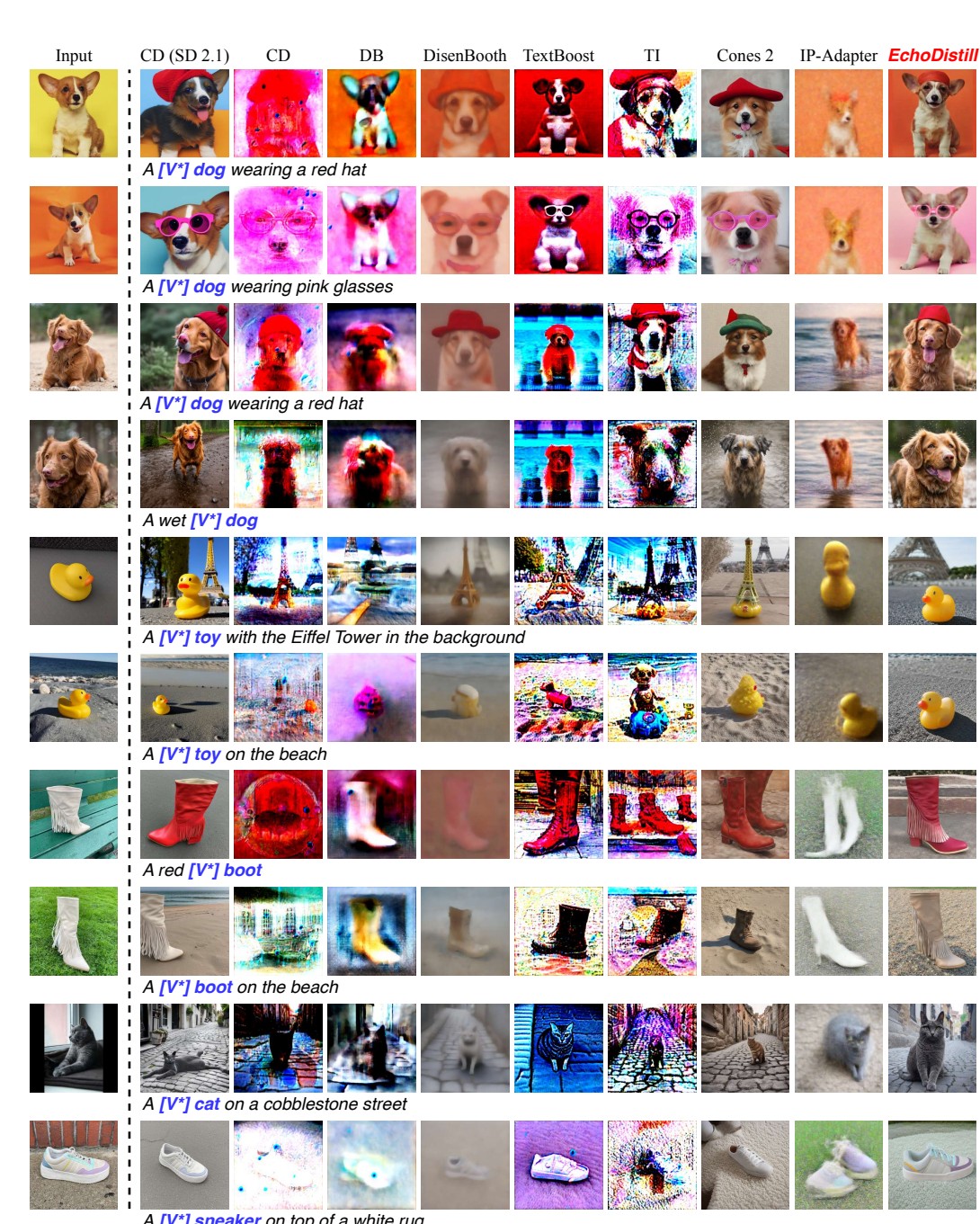

Figure 8: Our method *EchoDistill* (last column) compared with existing methods applied to the *1-SDP* setup with SDTurbo (Sauer et al., 2024) as the one-step diffusion backbone. One representative concept image is shown on the left-most column. (Part 3)

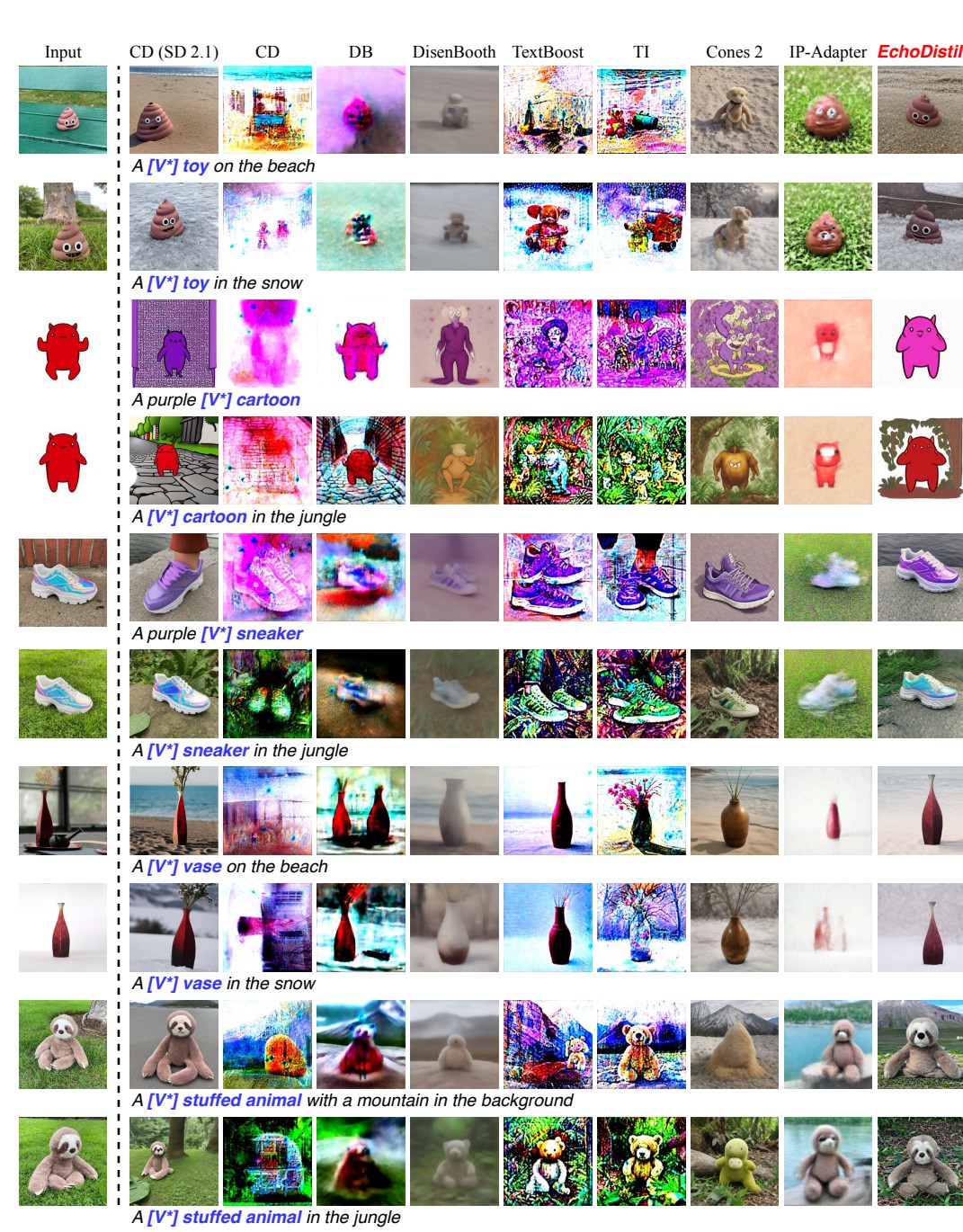

Figure 9: Our method *EchoDistill* (last column) compared with existing methods applied to the *1-SDP* setup with SDTurbo (Sauer et al., 2024) as the one-step diffusion backbone. One representative concept image is shown on the left-most column. (Part 4)

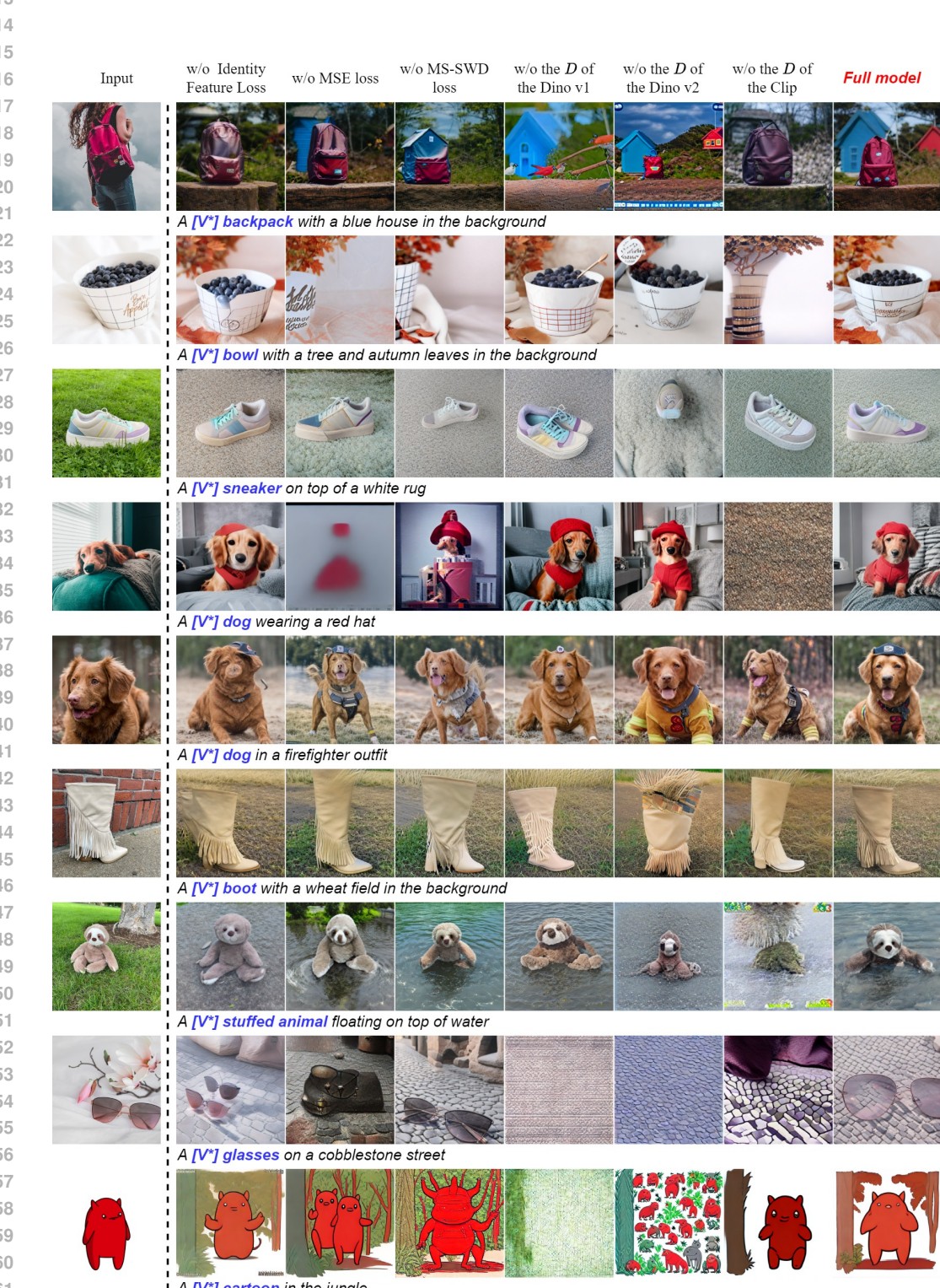

Figure 10: Qualitative results of the extended ablation study. $D$ denotes the discriminator.

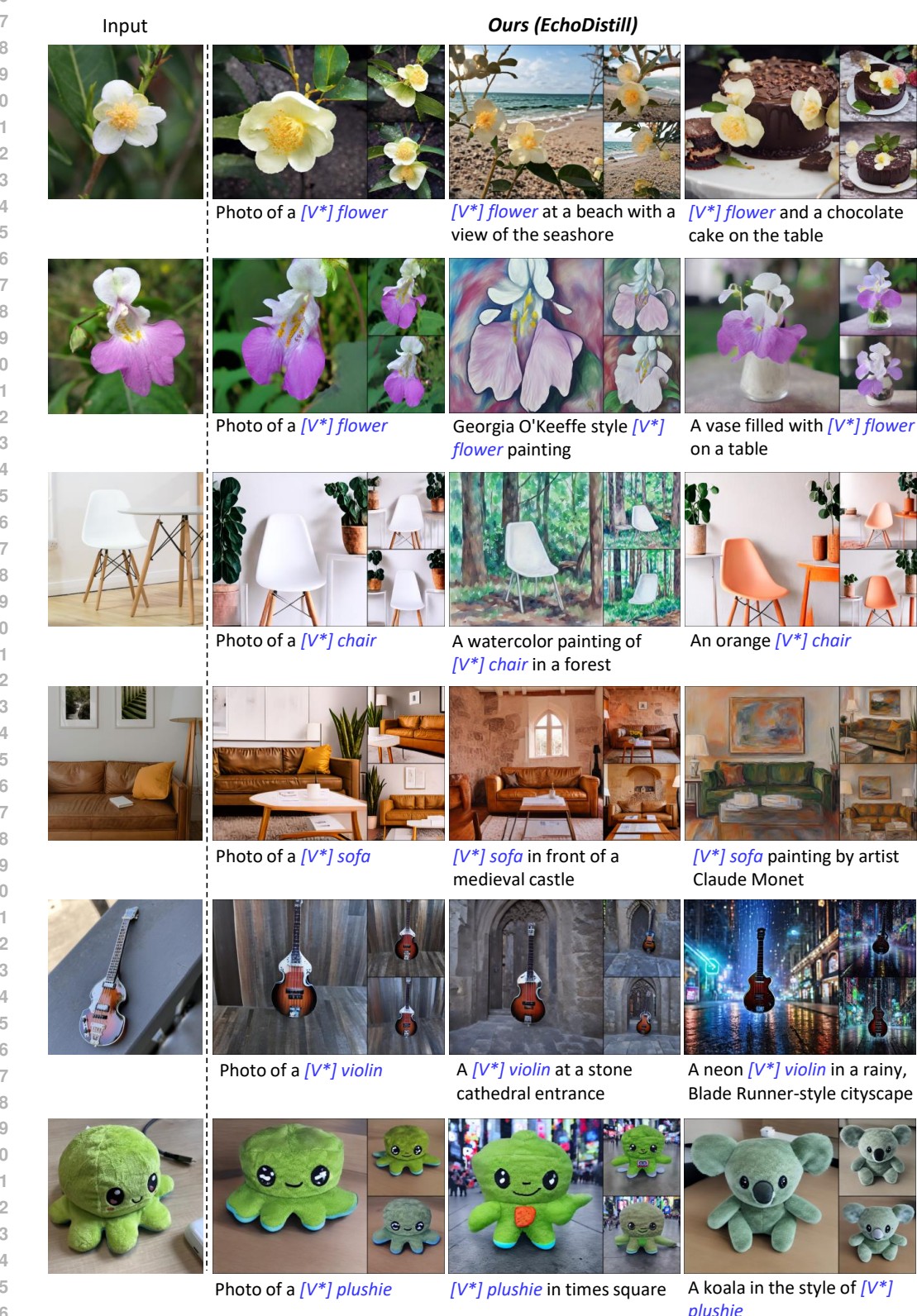

Figure 11: Qualitative results of *EchoDistill* on the CustomConcept101 dataset. Our method demonstrates strong generalization across a variety of concepts and prompt styles. (Part 1)

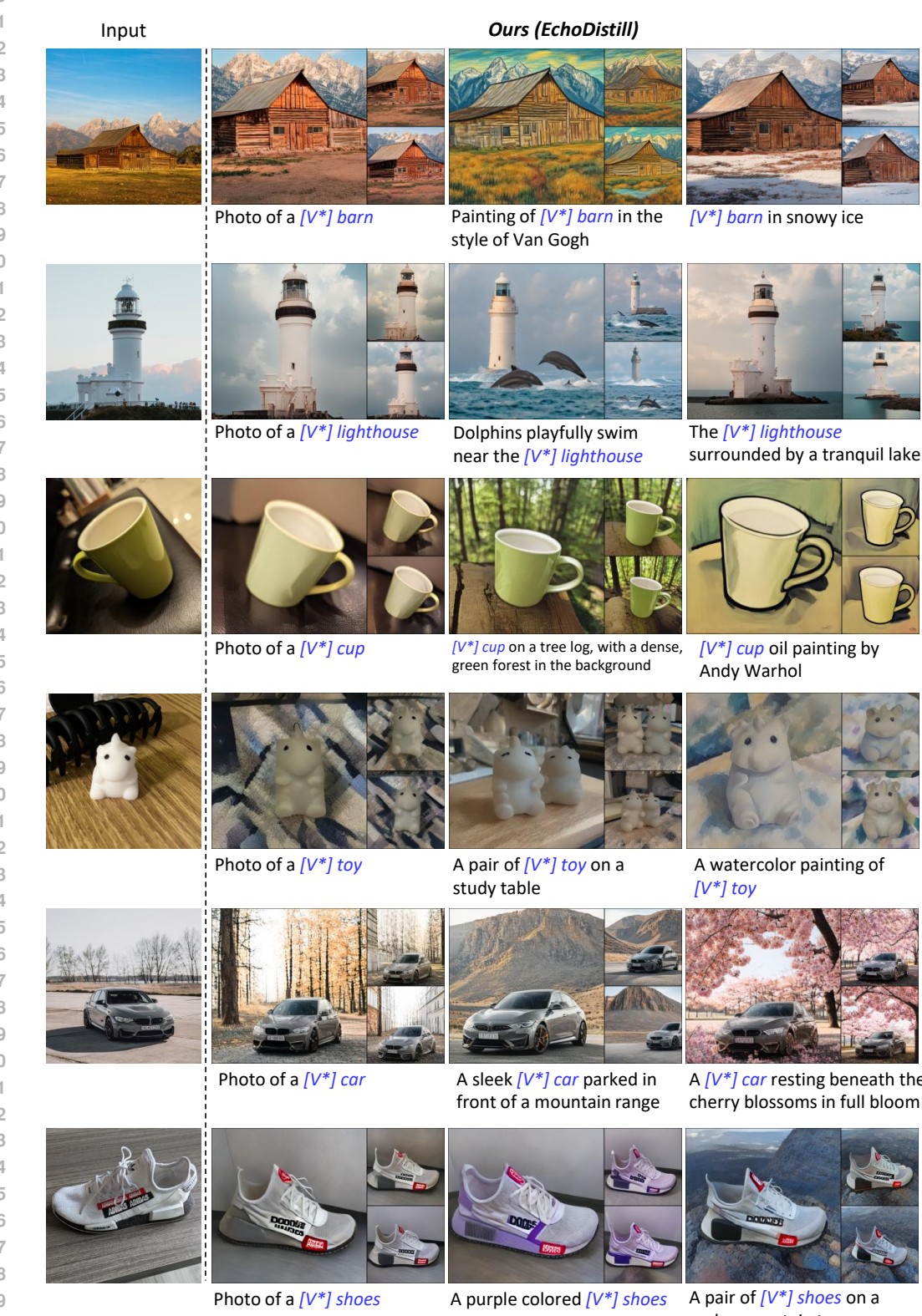

Figure 12: Qualitative results of *EchoDistill* on the CustomConcept101 dataset. Our method demonstrates strong generalization across a variety of concepts and prompt styles. (Part 2).

