# OpenReview forum: "EchoDistill: Bidirectional Concept Distillation for One-Step Diffusion Personalization"
_ICLR.cc/2026/Conference — ICLR 2026 Conference Withdrawn Submission_

### Official Review · Reviewer_Q97E · 2025-10-29

**Soundness:** 3
**Presentation:** 3
**Contribution:** 2
**Rating:** 4
**Confidence:** 3

**Summary:**

The paper introduces one-step diffusion personalization (1‑SDP) and proposes EchoDistill, a bidirectional teacher–student framework for fast text‑to‑image personalization. A multi‑step diffusion teacher (SD‑2.1) and a one‑step student (SD‑Turbo) are trained end‑to‑end with a shared text encoder, while the student receives (i) alignment losses (IPA/CLIP identity, MSE, MS‑SWD) to match the teacher's denoised outputs and (ii) an ensemble‑discriminator adversarial loss (CLIP, DINOv1/v2 features). An echoing stage then uses high‑quality student samples to further refine both models. On DreamBench (30 concepts), EchoDistill improves one‑step personalization over strong baselines in CLIP‑I and DINO similarity and remains competitive in CLIP‑T, with ablations supporting each component.

**Strengths:**

- Clear articulation of 1‑SDP and its challenges (student inadaptability, inefficiency, teacher unreliability).

- Shared text encoder plus teacher‑aligned and adversarial objectives; echoing stage is conceptually neat and empirically helpful.

- Highest CLIP‑I (0.783) and DINO (0.637) among compared methods while keeping CLIP‑T competitive.

- Component ablations, few‑step generalization without retraining, and backbone swap (Hyper‑SD1.5) provide useful evidence.

- Trains on a single A40 with modest hyper-parameters; adapts only K/V projections (Custom Diffusion style), which is parameter‑efficient.

**Weaknesses:**

- Adversarial discriminators use CLIP/DINO features; evaluation also reports CLIP‑I and DINO, risking optimistic bias. A metric independent of training features (e.g., DreamSim, human evaluation) would strengthen claims.

- Results center on DreamBench and SD-Turbo; broader datasets (e.g., faces, multi‑concept compositions) and more one‑step/few‑step backbones would improve generality.

- Using student samples as "real" may entrench artifacts or reduce prompt adherence; indeed CLIP‑T drops slightly after echo for the student. More safeguards/diagnostics are needed.

- IP‑Adapter is adapted via TCD due to incompatibility; stronger one‑step personalization baselines or teacher‑first pipelines should appear in the main text (not only the appendix).

- Missing training time per concept, number of steps/epochs, and prompt sets; limited discussion of failure modes and robustness to prompt variations.

**Questions:**

- Can you report metrics that do not reuse CLIP/DINO features (e.g., DreamSim or human preference) and analyze whether gains persist?

- How many echo rounds are used? Any safeguards against drift (e.g., mixing proportions of real vs. student data, confidence filtering, or consistency checks)? Show curves of CLIP‑T/CLIP‑I/DINO across echo iterations.

- What are training times per concept and wall‑clock comparisons to teacher‑first pipelines and other personalization methods? Include student inference latency vs. multi‑step teacher.

- Can you evaluate on additional datasets (e.g., human identity sets) and on multi‑concept composition prompts to assess scalability beyond single‑subject DreamBench?

- Provide sensitivity to the timestep weight c(t), discriminator backbones (e.g., removing CLIP vs. DINO), and the fraction of K/V parameters tuned. Also test echoing with a fixed teacher to isolate student‑only benefits.

- CLIP‑T slightly degrades for the student after echo (Table 3). Can you quantify text‑condition faithfulness with prompts requiring attribute/scene changes and discuss mitigation?

- Will you release code, prompts, and seeds for Figs. 3–4 and Tables 1–3 to enable verification?

---

### Official Review · Reviewer_3iZa · 2025-10-31

**Soundness:** 2
**Presentation:** 3
**Contribution:** 2
**Rating:** 4
**Confidence:** 4

**Summary:**

This paper focuses on one-step diffusion for image personalization. The main motivation is that existing acceleration methods often fail to capture fine-grained textural details, leading to reduced image fidelity and poor alignment between the edited samples and the reference images. The proposed method, EchDistill, performs knowledge distillation from a teacher model to a student model, followed by refinement of the teacher through feedback from the student. With this additional echoing stage, the student model effectively enhances the quality of the teacher model.

**Strengths:**

1. This paper focuses on one-step generation, aiming to address the issues of low-fidelity outputs and poor alignment between the generated content and the text instructions. The motivation is clear and well justified.

2. The introduction of an alignment loss to mitigate the misalignment problem is reasonable and well-motivated.

3. The echoing mechanism replaces real images with those generated by the student model, thereby forcing the teacher model to reconstruct samples from the student’s outputs. This strategy makes sense as it provides feedback from the student model to further guide the teacher in refining its concept learning process.

**Weaknesses:**

1. Overall, the alignment loss in Eq. (5) is designed to improve the consistency between the student and teacher models. However, it is unclear why the three loss terms (Eqs. 2–4) are chosen to work together, or how they complement one another. A more detailed explanation of their respective roles and interactions would be helpful.

2. The echoing stage is also not clearly explained. While its motivation (to use the student model’s output to guide the teacher model’s reconstruction) is understandable, it remains unclear how the degree of reconstruction is controlled to prevent the model from converging to undesirable modes. For instance, it's not clear how to balance the reconstruction of real image and the generated samples from the student model? Additionally, it would be useful to clarify whether the discriminator term (Eq. 7) serves a similar or overlapping purpose with the echoing mechanism.

**Questions:**

1. According to Table 8 in the supplementary material, the adversarial loss makes a significant contribution, which is reasonable. However, further analysis is needed to explain the differences among the three foundation models used as feature extractors, e.g. DINO V1, DINO V2, and CLIP. In particular, it would be valuable to discuss what inherent characteristics of these models lead to their differing effectiveness when serving as discriminators.

2. Although the ablation study on the three components of the alignment loss is presented in the supplementary material, the paper does not analyze why these specific terms were chosen or how they complement each other in contributing to the overall alignment objective.

3. What are the potential limitations of the proposed solution? A brief discussion would help clarify the scope and generalizability and robustness of the method.

---

### Official Review · Reviewer_Tmc3 · 2025-10-31

**Soundness:** 2
**Presentation:** 2
**Contribution:** 2
**Rating:** 4
**Confidence:** 4

**Summary:**

The authors of the paper propose to solve the personalization of single-step image generation diffusion models. Existing baseline personalization approaches do not apply to some of the one-step generation models; hence, they propose a distillation mechanism to personalize one-step models. They design a method of a single training process, to learn student and teacher at the same time, with an additional 2 groups of losses: adversarial and alignment losses.

**Strengths:**

1. The authors tackle a challenging problem of personalizing single-step image generation models and present some results.
2. Joint distillation mechanism for concept personalization of diffusion models.

**Weaknesses:**

1. Only baseline and old methods were used as competitors (DreamBooth / DreamBench). Many other advanced and improved personalization methods are ignored in the comparisons, such as NeTI [1], PALP [2], and others.
2. There is some sort of exaggeration of Table 1’s value. Often referred to as a justification or validation for some claims.
E.g. Where do the results of Table 1 prove that student models sometimes outperform teachers’ results with visual qualities (314-316)?
3. A sophisticated method design with multiple losses and multiple models involved in a single process.
4. Poor fidelity in the qualitative results of Fig. 3. E.g. first image’s object and the EchoDistill results have different cover.
5. Reimplementation of some of the competitor methods, IP-Adapter, and multiple models (mentioned in Table 4) questions the best performance of them. They may not produce the best results that the authors achieved in their original papers.


[1] A Neural Space-Time Representation for Text-to-Image Personalization, Yuval Alaluf et al.

[2] PALP: Prompt Aligned Personalization of Text-to-Image Models, Moab Arar et al.

**Questions:**

1. What is "y" in Eq. 1? It does not contain an explanation of notations.
2. How did the entire training process, which has an SDXLTurbo as a student, another multiple-step image generation model (at least another SDXLTurbo), 3 discriminator models (enlarged CLIP, DINOv1, and DINOv2), fit into a single A40 GPU with 48GB memory at the same time for joint training in a single process (as the authors claim in their paper)?

---

### Official Review · Reviewer_nupS · 2025-11-05

**Soundness:** 3
**Presentation:** 3
**Contribution:** 2
**Rating:** 4
**Confidence:** 4

**Summary:**

This paper presents EchoDistill, a bidirectional concept distillation framework for one-step diffusion personalization (1-SDP). It jointly trains a multi-step teacher and a one-step student with a shared text encoder to ensure semantic consistency. After the teacher distills concept knowledge to the student, the student “echoes” feedback to refine the teacher using its fast generation ability. This bidirectional learning improves both models—enhancing personalization accuracy and generation quality. Experiments on DreamBench show that EchoDistill outperforms existing personalization methods in the one-step diffusion setting.

**Strengths:**

- The concept of bidirectional learning is highly inspiring: the design where the student model provides feedback to refine the teacher opens up a new perspective for collaborative training in diffusion-model personalization.

- The empirical results show that on metrics such as CLIP-I and DINO the proposed framework significantly outperforms existing one-step or multi-step personalization methods; notably, it even surpasses the multi-step-based method Custom Diffusion, which demonstrates the effectiveness and generalization ability of the framework.

**Weaknesses:**

- Insufficient motivation for introducing the Echo mechanism: The authors claim that “the teacher model may struggle to accurately learn certain visual concepts” and “we exploit the student model’s rapid generation capability to provide constructive feedback.” However, since the student’s generated samples are likely to be of low quality in the early training stage, it remains unclear why such imperfect feedback can effectively improve the teacher. The paper lacks quantitative or visual analyses to show how and to what extent the echo stage enhances either the teacher or the student model.

- Limited experimental scope: All experiments are conducted only on the DreamBench benchmark with the SD-Turbo one-step diffusion model. This narrow setup limits the generalizability of the conclusions. Validation on other datasets (e.g., Subject-Diffusion) or in multi-concept personalization scenarios would make the empirical evidence more convincing and demonstrate broader applicability of the proposed framework.

**Questions:**

Please refer to the weakness part.

---

### Note · Authors · 2025-11-13

I have read and agree with the venue's withdrawal policy on behalf of myself and my co-authors.